# Whole-Grain Intake in the Mediterranean Diet and a Low Protein to Carbohydrates Ratio Can Help to Reduce Mortality from Cardiovascular Disease, Slow Down the Progression of Aging, and to Improve Lifespan: A Review

**DOI:** 10.3390/nu13082540

**Published:** 2021-07-25

**Authors:** Cristiano Capurso

**Affiliations:** Department of Medical and Surgical Sciences, University of Foggia, Viale Pinto 1, 71122 Foggia, Italy; cristiano.capurso@unifg.it

**Keywords:** aging, frailty, lifespan, diet, carbohydrates, whole grain, protein

## Abstract

Increase in the aging population is a phenomenon all over the world. Maintaining good functional ability, good mental health, and cognitive function in the absence of severe disease and physical disability define successful aging. A healthy lifestyle in middle age predisposes successful aging. Longevity is the result of a multifactorial phenomenon, which involves feeding. Diets that emphasize fruit and vegetables, whole grains rather than refined grains, low-fat dairy, lean meats, fish, legumes, and nuts are inversely associated with mortality or to a lower risk of becoming frail among elderly subjects. A regular physical activity and a regular intake of whole grain derivatives together with the optimization of the protein/carbohydrate ratio in the diet, where the ratio is significantly less than 1 such as in the Mediterranean diet and the Okinawan diet, reduces the risk of developing aging-related diseases and increases healthy life expectancy. The purpose of our review was to analyze cohort and case-control studies that investigated the effects of cereals in the diet, especially whole grains and derivatives as well as the effects of a diet with a low protein–carbohydrate ratio on the progression of aging, mortality, and lifespan.

## 1. Introduction

According to the World Health Organization, population aging is a global phenomenon rapidly evolving worldwide. By 2030, the number of people aged 60 and over in the world is projected to grow from 901 million to 1.4 billion, or 56%. It is expected that by 2050, the global population of people over 65 will amount to about 2.1 billion people, more than double compared to 2015. In addition, it is estimated that by 2050, the over eighty-year-olds throughout the world will be around 434 million, or more than three times compared to 2015, when they reached 125 million. The rapid aging of the population can be observed above all in emerging economy countries. In fact, over the next 15 years, the elderly population will grow more rapidly in Latin America and the Caribbean with an expected increase of 71%, followed by Asia (66%), Africa (64%), Oceania (47%), North America (41%), and Europe (23%) [1]. This means that while European countries have had more than 150 years to adjust to an increase of up to 20% in the proportion of the population over 65, countries like Brazil, China, and India will have less than 20 years to adapt to a similar one. The population as of 1 January 2018 in the European Union (EU) was estimated to be 512.4 million. People over 65 years old amounted to 19.7%, an increase of 2.6% compared to 10 years earlier. The percentage of people aged over 80 is expected to at least double by 2100 to 14.6% of the entire EU population [2].

It is also true that many elderly people maintain good autonomy and live life with a good level of well-being. These subjects, despite the presence of one or more diseases, however, do not have serious illnesses or physical disabilities; they have good mental health, preserved cognitive functions, maintain a good level of physical activity levels and in some cases, are engaged in social and productive activities [3,4]. All these conditions define successful aging.

It is known that a healthy life in middle age predisposes successful success. This includes a healthy diet with adequate caloric intake to the state of health and physical activity, smoking cessation, and taking moderate amounts of alcohol, preferably with meals. The traditional Mediterranean diet (MD) is characterized by a high intake of foods of plant origin (fruit, vegetables, whole-meal bread, beans, nuts, and seeds) and fresh fruit; extra virgin olive oil is the main dietary source of fat.

Traditional MD has long been recognized as a highly healthy dietary pattern. High adherence to traditional MD leads to a significant reduction in mortality and a reduced risk of developing cardiovascular disease and cancer as well as a reduced risk of developing chronic disease and disability in later life. The main source of complex carbohydrates is made up of cereals and their derivatives (bread, pasta, rice); these provide 55–60% of the total caloric intake and are placed at the bottom of the food pyramid [5,6,7,8,9,10,11,12,13,14,15]. 

Another health diet model other than MD is the traditional Okinawan diet [16]. This is also characterized by a low overall caloric intake, high consumption of vegetables, high consumption of legumes (mainly soybeans), moderate consumption of fish, especially in coastal areas, in any case, by the low consumption of meat, especially lean pork. Characteristic of traditional Okinawa is also a low consumption of dairy products, a high intake of mono- and polyunsaturated fats, with a low omega 6:3 ratio, the consumption of low glycemic index carbohydrates with a high intake of fiber, and a moderate consumption of alcohol. Figure 1 compares the composition of the MD and the Okinawan diets. 

The purpose of our review was to analyze both cohort and case-control studies that investigated, on one hand, the effects of cereals, of whole grains (WG), and derivatives in the diet, on the other, the effects of a diet with low protein–carbohydrate ratio on aging progression, mortality, and lifespan.

## 2. Cereals

Cereals (from Ceres, the Roman goddess of crops and fields) have been the staple food for most people around the world since ancient times. Cereals, especially when consumed as WG [17], are a healthy source of carbohydrates, fiber, and bioactive peptides with anticancer, antioxidant, and antithrombotic effects [18]. In traditional MD [19], grains provide up to 47–50% of the daily calorie intake. The cereals and derivatives mainly consumed in MD are wheat, spelled, oats, rye, barley, and, to a lesser extent, rice and maize. Table 1 summarizes the nutritional properties of all the above cereals.

### 2.1. Wheat

Wheat (Triticum aestivum, Triticum durum) is a cereal of ancient culture, whose area of origin is located between the Mediterranean Sea, the Black Sea, and the Caspian Sea, and is currently cultivated all over the world [20]. Wheat has a protein content of 13–14%, higher than that of the other main cereals and staple foods; therefore, it is the main plant source of protein in human nutrition worldwide. A total of 100 g of wheat provides 327 calories; wheat is also an important source of dietary fiber, niacin, several B vitamins, and other dietary minerals. Furthermore, 75–80% of total wheat protein is made up of gluten [21].

#### 2.1.1. Starch and Protein

Starch, on average, is approximately 80% of the dry weight of the endosperm and consists of a mixture of two polymers, amylose and amylopectin, in a ratio of about 1:3. The protein content of wheat has wider variations than the starch content [22]. An analysis from the World Wheat Collection, after comparing 212,600 germplasm lines, showed a wide variability of the protein content, with a range from 7 to 22% of protein on dry weight [23]. Similarly, the result of the comparison analysis between 150 lines of wheat grown under the same agronomic conditions, as part of the HEALTHGRAIN program, highlighted a variation in the protein content of wheat from 12.9 to 19.9% with regard to wholemeal flours and from 10.3 to 19.0% for white flours [24] More than half of the total protein content of the wheat grain, as already stated above, is made up of gluten, in a measure directly proportional to the total protein content [25]. 

#### 2.1.2. Wheat Fibers and Cell Wall Polysaccharides

According to the 2009 Codex definition [26], dietary fiber (DF) is a “…carbohydrate polymer with a degree of polymerization (DP) not lower than 3, which are neither digested nor absorbed in the small intestine…” 

The European Commission under Commission Directive 2008/100/EC [27], subsequently established under Regulation (EU) No. 1169/2011 of The European Parliament and of The Council [28], further defines DF. In this definition, all carbohydrates with a degree of polymerization (DP) ≥3 can be included in dietary fiber; of these, the most common in cereals are fructo-oligosaccharides.

Whole wheat is among the main sources of DF and mainly comprises non-starch polysaccharides (NSPs), which are derived from the cell walls. Most of the fibers are removed during grinding, as refined flour has an extremely low amount of fiber. The amount of fiber in whole wheat varies from 12 to 15% of the dry weight, mainly concentrated in the bran. The most common fiber of wheat bran, equal to about 70%, is arabinoxylan (Figure 2); this is composed of hemicellulose, and β-glucan (20%) as well as a small amount of cellulose (2%) and glucomannan (7%) [29]. Bran obtained from grinding includes a set of compounds that comprise up to 45–50% of cell wall material [30]. The pericarp is the main component and is composed of about 30% of cellulose, about 60% of arabinoxylan, and about 12% of lignin [31].

#### 2.1.3. Antioxidant Components and B Vitamins in Wheat

The grain of wheat contains numerous antioxidants, mainly concentrated in the bran and germ, parts absent in refined white wheat flour. The main antioxidants in the wheat grain are terpenoids (including vitamin E) and phenolic acids [21]. In the wheat grain, phenolic acids are mostly derivatives of hydroxycinnamic acid. In particular, these are dehydrodimers and dehydrotrimers of ferulic acid and synapic and p-coumaric acids [32]. In the outer layer of the bran, we find most of the phenolic acids, mostly bound through ester bonds, to the structural components of the cell wall. The highest shares of antioxidants are found in the outermost layer of the endosperm (i.e., the aleurone). Therefore, the antioxidant properties (i.e., the presence of relevant quantities of phenolic compounds) are directly correlated to the aleurone content of the wheat grain [33]. Among the polyphenols of wheat and other cereals, ferulic acid is the predominant. Other classes of antioxidants contained in wheat bran are flavonoids, carotenoids (mainly lutein), and lignans [34,35].

Wheat is an important source of the so-called “methyl donors”, important cofactors in the methylation process, necessary for the synthesis of dopamine and serotonin as well as for the biosynthesis of melatonin and coenzyme Q10. The main component is betaine glycine, therefore, in smaller quantities, it is choline (precursor of betaine) and trigonellin (a structural analogue of betaine and choline). Concerning B group vitamins, wheat is a good source of thiamin (B1), riboflavin (B2), niacin (B3), pyridoxine (B6), and folate (B9) [21].

#### 2.1.4. Health Effects

The health effects of wheat are due to the high content of numerous nutrients and fibers as well as proteins and minerals. Wheat, if consumed as whole wheat, is recommended in several daily portions in the nutrition of both children and adults in quantities equal to about one third of the total diet. For example, whole wheat is a common component found in breakfast cereals and is associated with a reduced risk for various pathologies. Thanks also to the high intake of insoluble fiber, whole wheat in the diet contributes to reducing the risk of coronary heart disease [CHD], stroke, cancer, and type 2 diabetes mellitus as well as helping to reduce mortality due to all causes [36,37].

### 2.2. Rye

Rye (Secale cereale) is part of the Graminaceae family (Triticeae), and is similar to barley (genus Hordeum) and wheat (Triticum). Rye is used for the production of flour, bread, crispbreads, beer, whiskey, vodka; it is also used as forage for animals [20].

#### 2.2.1. Nutrition Properties

A 100 g serving of rye contains 338 calories and consists of carbohydrates (28%), proteins (20%), dietary fiber (54%), niacin (27%), pantothenic acid (29%), riboflavin (19%), thiamine (26%), vitamin B6 (23%), and minerals. [21].

Compared to wheat flour, rye flour has a lower gluten content, being rich in gliadin but low in glutenin. Although in small quantities, the gluten content makes rye a cereal unsuitable for consumption by people with celiac disease, non-celiac gluten sensitivity, or wheat allergy.

#### 2.2.2. Health Effects

Thanks to the high content of non-cellulosic polysaccharides, rye is an excellent source of fiber, with an exceptionally high capacity to bind water, and which therefore quickly gives a feeling of fullness and satiety. For this reason, rye bread is a valuable aid in the weight loss diet.

#### 2.2.3. Rye Bread and Glucose Metabolism

Juntunen et al. [38] evaluated, in a sample of 20 healthy, non-diabetic, postmenopausal women, the effect on insulin response after intake of refined wheat bread, endosperm rye bread, traditional whole-meal rye bread, and high-fiber rye bread. They measured blood glucose and insulinemia, glucose-dependent insulinotropic polypeptide (GIP), and glucagon-like peptide 1 (GLP-1). All these markers of insulin response were measured in blood samples taken at fasting (time 0) and respectively after 15, 30, 45, 60, 90, 120, 150 and 180 min from the consumption of the different types of bread. The authors demonstrated that post-prandial blood glucose values after the consumption of rye bread were not significantly different from the values measured after the consumption of refined white wheat bread. In contrast, the blood values of insulin, GIP, and C-peptide after the consumption of rye bread were significantly lower than the values obtained after the consumption of wheat bread (*p* < 0.001). Furthermore, plasma GLP-1 values after consumption of rye bread were not significantly different from those obtained after consumption of the other breads, except at 150 and 180 min (*p* = 0.012). The authors also demonstrated that the lower insulin response after eating rye bread cannot simply be explained by the higher amount of fiber contained in rye bread. Micrographic examination revealed differences in the structure of refined wheat bread, rye endosperm bread, high fiber rye bread, and traditional rye bread. For example, in wheat bread, gluten proteins formed a continuous matrix in which the starch grains were dispersed. On the other hand, in the rye bread, the starch grains were more swollen and the amylose was partially leached. The starch granules were well packaged and formed a continuous matrix. It was clear, therefore, that the softness and porosity of refined wheat bread and the hardness of rye bread were based on these differences in their structure.

Nordlund et al. [39] subsequently confirmed these data. They analyzed the mechanical, structural, and biochemical properties of various types of rye and wheat bread as well as the particle size of the breads after gastric digestion in in vitro and in vivo glycemic and insulin responses on a sample of 29 volunteers. Therefore, 10 different types of bread from ten different flours were packaged, with 10 different characteristics of composition and consistency, namely: refined wheat, whole rye, whole rye (commercial), whole rye + bran, refined rye, refined rye (flat), refined rye + gluten (flat), rye/whole wheat, wheat/whole wheat, and refined wheat + fermented bran. A sourdough baking process was used for baking rye breads, while a straight dough baking process was used for baking wheat breads. Upon microscopic observation, both 100% wholemeal rye flour bread and sourdough refined rye flour bread had a higher number of digestive particles larger than 2 or 3 mm in size, meaning that they appeared less “disintegrated” “compared to wheat flour bread. Microstructural examination of the digestive particles of sourdough rye bread also showed more aggregated and less degraded starch granules than refined wheat bread. The postprandial insulin response produced from 100% rye flour bread by the sourdough method was significantly lower than the insulin response produced by refined wheat flour bread (*p* = 0.001). From principal component analysis (PCA), the authors confirmed that the insulin response was inversely related to the larger digestive particle size obtained after in vitro digestion, the number of soluble fibers, and the sourdough process. That is, the larger starch particles obtained after gastric digestion of bread from wholemeal rye flour were associated with a reduced postprandial insulin response. This mechanism, likely in synergy with fiber and WG, explains the reduction in the risk of diabetes obtained with the consumption of rye bread in the diet.

More recently, Rojas-Bonzi et al. [40] conducted a study on pigs with a catheterized portal vein fed on wheat bread and wholemeal rye bread to analyze the kinetics of the in vitro digestion of breads by varying the dietary fiber content and composition, thus comparing the results obtained with the data of a previous in vivo study [41]. Five varieties of bread were analyzed: white wheat bread (WWB), whole grain rye bread (WRB), and whole grain rye bread with kernels (WRBK), which were commercial breads; in addition, two varieties of experimental breads (i.e., specially prepared for the study: concentrated wheat Arabinoxylan (AXB) and concentrated wheat β-glucan (BGB)). As expected, WWB had the highest total starch content (711 g/kg dry matter, DM), while the starch content was lowest in all high DF content breads (588, 608, 514, 612 g/kg DM, respectively). Total DF was low in the WWB (77 g/kg DM) and high in all high DF breads (209, 220, 212, 199 g/kg DM, respectively). Total DFs were lowest in WWB (77 g/kg DM) and highest in all high-DF breads (209, 220, 212, 199 g/kg DM, respectively). Of course, the characteristics of the total and soluble DFs varied considerably between the loaves. The BGB had a high content of total and soluble β-glucan (52 and 40 g/kg DM), while the WRB, WRBK, and AXB had a high content of total and soluble arabinoxylan (76 and 36, 77 and 37, 78 and 66 g/kg DM, respectively). The highest percentage value of starch hydrolysis in vitro was observed from time 0 and within the first 5 min and subsequently decreased. The highest rate of hydrolysis during the first 5 min was observed in WWB (13.9% starch/min), followed by WRB (10.4% starch/min), WRBK (8.7% starch/min), and finally from AXB and BGB (7.4–8.5% starch/min). In order to be able to compare the data obtained in vitro with the in vivo data, the measurement of portal glucose values was reported by the authors as a percentage of hydrolyzed starch (absorbed starch) per 100 g of dry starch (ingested starch). After the first 15 min, the highest values were observed in the WWB, the lowest values for the WRB and WRBK, and intermediate values for the AXB and BGB (*p* < 0.05). The authors explained the extremely high rate of hydrolysis of the WWB with a porous physical structure of white wheat flour, which makes the readily degradable bread. The quantity of DF, both naturally present in the cell walls (WRB, WRBK) and added (AXB, BGB), delays its digestion in vitro, extending the hydrolysis time in the first 5 min. The greatest effect was observed in the BGB, probably due to the increased viscosity of the BGB compared to other types of bread. The reduced in vitro digestion rate within the first 5 min of arabinoxylan compared to b-glucan is due to its more branched structure. Arabinoxylan is also less sensitive to the change in acidity during the passage from the stomach to the small intestine, unlike b-glucan. The authors therefore confirmed the results already obtained by Juntunen et al. [38], or that the processing of white wheat bread gives it a more porous structure to rye bread, which has a more compact structure. The inclusion of unrefined grains in bread has also been proven to be an efficient way to regulate starch hydrolysis: the insoluble fibrous network surrounds the starch, forming a real physical barrier against amylases, limiting its gelatinization. The viscous nature of soluble DFs further increases the viscosity of the digestive bolus, limiting its diffusion and delaying the absorption of glucose through intestinal cells.

### 2.3. Spelt (Triticum Spelta)

Spelt (Triticum spelta), is a species of wheat that has been cultivated since ancient times. It originated as a natural hybridization of a domesticated tetraploid wheat and a wild goat grass Aegilops tauschii. 

In the twentieth century, spelt was almost completely replaced by wheat flour bread, but it has become popular again in recent years, thanks to the spread of organic agriculture. Spelt is very disease resistant and also grows in poor growing conditions such as wet and cold soils or at high altitudes, and requires less fertilizer. Furthermore, it does not require any chemical treatment of the hulled seeds used for sowing, thanks to the protection provided by the hull [20].

#### Nutrients

A 100 g of raw spelt provides 338 calories. It is composed of about 70% carbohydrates, of which 11% is dietary fiber, and is low in fat. Spelt has a good protein content; it is also a terrific source of dietary fiber, B vitamins including niacin and of a wide variety of dietary minerals including manganese and phosphorus [21]. The comparison between nine samples of hulled spelt and five of soft winter wheat [42] showed a higher average quantity of total lipids and unsaturated fatty acids, with a lower tocopherol content, both in whole spelt and in spelt from grinds, compared to wheat. This suggests that the higher lipid content of spelt might not be related to a higher proportion of germs. The proportions of flour and bran after grinding were similar in spelt and wheat; the content of ash, copper, iron, zinc, magnesium, and phosphorus was higher in the samples of spelt, particularly in fine bran rich in aleurone and in the coarse bran. The phosphorus content was higher, while the phytic acid content was lower in spelt than in fine wheat bran. This could suggest that spelt has either a higher endogenous phytase activity or a lower phytic acid content than wheat. 

Compared to hard red winter wheat, spelt has lower insoluble polymeric proteins, which contribute to the swelling capacity of the gluten. Spelt also has higher gliadins, which have the opposite effects, and higher values of soluble polymeric proteins. It follows that the gluten in spelt is less elastic and more extensible than wheat gluten, resulting in the typical weaker spelt dough [43].

### 2.4. Oats

Oat (Avena sativa, the best known species of the Avena genus), unlike other varieties of cereals and pseudocereals, is cultivated for their seed, known by the same name, usually in the plural. Oats are commonly eaten rolled or ground as oatmeal or as fine oatmeal and consumed primarily as porridge, but are also used as an ingredient for making cakes, cookies, and bread. Oats are also an ingredient in breakfast cereals, particularly in muesli. In the United Kingdom, oats are used for the production of beer. A popular refreshment throughout Latin America is a characteristic cold, sweet drink made from ground oats and milk [20].

#### 2.4.1. Nutrients

A 100 g of oats provide 389 calories. Oats are made up of about 66% carbohydrates, 11% dietary fiber, 4% beta-glucans, 7% fat, and 17% protein. Oats are also an excellent source of B vitamins and minerals, particularly manganese [21].

After corn, oats have the highest lipid content of most other cereals of over 10% compared to 2–3% for wheat. Furthermore, oats are the only cereal containing a globulin, avenaline, as the main storage protein (around 80%). Compared to gluten, zein, and prolamins, the most typical cereal proteins, globulins, are characterized by their solubility in diluted saline solution. Avenin, a prolamine, is the minor protein of oats. In nutritional qualities, oat proteins are almost equivalent to soy proteins, which in turn are equivalent in nutritional quality to proteins in meat, milk, and eggs, according to research by the World Health Organization. A skinless oat grain (semolina) has a protein content ranging from 12 to 24%, the highest among cereals. Some pure oat cultivars (oats not contaminated by other gluten-containing grains) can be a safe food in a gluten-free diet, which requires knowledge of the varieties of oats used in foods. Oats contain about 11% fiber, most of which is composed of b-glucans, indigestible polysaccharides found naturally in cereals as well as in barley, yeast, bacteria, algae, and fungi [14,20]. Oats, particularly the more “ancient” varieties, contain more soluble fibers than common western varieties, which induce a slowdown in digestion with a consequent greater feeling of satiety and reduced appetite [44,45]. 

It has been shown that dietary benefits from whole oats are associated with an improved control of cardio-metabolic risk factors by reducing blood lipids and blood glucose. Eating oat-based foods, either as whole grains or as bread, porridge, or soaking oats in milk, has been shown to allow for better glycemic control [46,47,48,49,50,51].

#### 2.4.2. Oat Beta-Glucan

Oat beta-glucan is made up of mixed-bonded polysaccharides. This means that the bonds between the D-glucose or D-glucopyranosyl units are beta-1, 3 or beta-1, 4 bonds. This type of beta-glucan is also defined as a mixed bond (1 → 3), (1 → 4)-beta-D-glucan (Figure 3). These bonds (1 → 3) break the uniform structure of the beta-D-glucan molecule and make it soluble and flexible. In comparison, the cellulose indigestible polysaccharide, which is also a beta-glucan, is not soluble because of its (1 → 4)-beta-D-bonds. The percentages of beta-glucan vary in the various products based on whole oats such as oat bran (range 5.5–23.0%), oat flakes (about 4%), and oat flour integral (about 4%). Oats also contain some insoluble fibers including lignin, cellulose, and hemicellulose [20]. Beta-glucans are known to have cholesterol-lowering properties as they increase the excretion of bile acids, with a consequent reduction in blood cholesterol [52]. This cholesterol-lowering effect of beta-glucans has allowed oats to be classified as a health food [53].

### 2.5. Rice

Rice is the seed of the monocotyledonous flowering plants Oryza glaberrima (African rice) or Oryza sativa (Asian rice). It is the most consumed cereal by the human population in the world and is the basis of Asian cuisine. It is the staple food for about half of the world’s population and is grown in almost every country in the world. It is the agricultural product with the highest world production (741.5 million tons recorded in 2014), after sugar cane (1.9 billion tons) and corn (1.0 billion tons). There are many varieties of rice, and culinary preferences tend to vary regionally.

#### Nutrients

The nutritional value of rice depends on several factors. First of all, it varies according to the rice strain, that is white rice, brown rice, red rice, or black rice, which have a different percentage of distribution in different regions of the world [54]. After that, the nutritional value of rice depends on the nutrient quality of the soil in which it is grown, if and how it is polished or processed, and if and how it is enriched and how it is prepared before consumption [55].

A 100 g serving of unenriched white rice provides an average of 360 calories, distributed between carbohydrates, proteins, fats, and fibers. Rice is also a good source of B vitamins and several dietary minerals including manganese. Raw white rice contains 66% carbohydrates, mostly starch, 11% dietary fibers, 4% beta-glucans, 7% fats, and 17% proteins. Cooked unenriched white rice is composed of 68% water, 28% carbohydrates, 13% protein, and fat in minimal quantity (less than 1%). Cooked short-grain white rice provides the same food energy and contains moderate amounts of B vitamins, iron, and manganese (10–17% of daily value, DV) per 100-g serving [21].

Starch and proteins, as main components of rice grains, accumulate in specific organelles called amyloplasts and protein bodies, respectively, in the endosperm cells and in the aleurone layer. Endosperm cells contain many amyloplasts with multiple starch grains and protein bodies with glutellin (protein body II) and prolamine (protein body I), which are storage proteins. On the other hand, the cells in the aleurone layer contain another type of protein body called grain aleurone, with non-storage proteins and small amyloplasts. The protein content of rice grains is of course lower than meat (15–25%) and cheese (20%), but is higher than dairy milk (3.3%) and yoghurt (4.3%). About 6–7% of polished rice and about 13% of rice bran is protein [56].

Amino acid score, in combination with protein digestibility, which refers to how well a given protein is digested, is the method used to determine if a protein is complete (i.e., whether it contains an adequate proportion of each of the nine essential amino acids necessary in the human diet). Together with the amino acid score, the digestibility of proteins determines the values for Protein Digestibility-Corrected Amino Acid Score (PDCAAS) and Digestible Indispensable Amino Acid Score (DIAAS). DIAAS was proposed in March 2013 by the FAO to replace the PDCAAS. DIAAS provides a more accurate measure of the number of amino acids absorbed by the body or the contribution of the protein to the needs of amino acids and nitrogen in humans, as it estimates the digestibility of amino acids at the end of the small intestine. PDCAAS, already adopted by the FAO in 1993 as a method for determining the quality of proteins is based on an estimate of crude protein digestibility determined over the total digestive tract, and values stated using this method generally overestimate the number of amino acids absorbed [57]. Compared with casein, which has a DIAAS of 101, rice has a DIASS of 47, whereas wheat has a DIASS of 48, oat has a DIASS of 57, and corn (Maize) has a DIASS of 36 [58]. If instead we take into consideration the PDCAAS, rice bran protein has a PDCAAS of 0.90, whereas casein has a PDCASS of 1.00, and rice endosperm protein has a PDCAAS of 0.63 [59]

### 2.6. Maize (Corn)

Maize, also known as corn, is a large grass plant already domesticated by the native populations of Mexico about 10,000 years ago. The word corn derives from the term “mahiz”, with which the indigenous Taino people of the Caribbean and Florida called the plant, later transliterated into Spanish. In the United States, Canada, Australia, and New Zealand, the term mainly refers to maize with the term “corn”, derived from the shortening of the expression “Indian corn”, which mainly refers to maize, which is the staple cereal of Native Americans [20].

#### 2.6.1. Nutrients

A 100 g serving of uncooked corn kernels provide 86 calories; it contains 3.27 g of proteins, 18.7 g of carbohydrates, 2 g of fibers, 6.26 g of sugars, and 1.35 g of fats, of which 26% of saturated fatty acids, 39% of polyunsaturated fatty acids, and 35% of monounsaturated fatty acids. Raw maize is a good source of group B vitamins, particularly niacin (11% of DV), riboflavin (4% of DV), thiamine (13% of DV), and vitamin B6 (7% of DV). Raw maize is also a good source of several dietary minerals, especially copper (6% of DV), iron (3% of DV), magnesium (9% of DV), manganese (7% of DV), phosphorus (13% of DV), potassium (6% of DV), zinc (4% of DV), selenium (1% of DV), and sodium (1% of DV) [21].

#### 2.6.2. Maize Oil 

Corn oil (corn oil, CO) is obtained by extraction from the corn germ. It is mainly used in the kitchen, thanks to its high smoking temperature, which makes corn oil suitable for frying. It is also a staple ingredient in margarine production. It is also used as an excipient in the pharmaceutical industry [20].

A total of 100 g of maize oil contains 13% of saturated fatty acids, of which 82% is palmitic acid (C 16:0) and 14% is stearic acid (C 18:0); 28% of monounsaturated fatty acids, of which 99% is oleic acid (C 18:1); and 55% of polyunsaturated fatty acids, of which 98% is linoleic acid (C 18:2), and 2% is omega-3 linolenic acid (C 18:3) [21,60].

#### 2.6.3. Corn Oil vs. Extra-Virgin Olive Oil

Unlike CO, whose production takes place through the solvent extraction of the oil from the grain after the separation of the corn germ with fragmentation or centrifugation, the production of olive oil takes place essentially by mechanical pressing of the drupe. A 100 g serving of extra virgin olive oil (EVOO) provides 884 calories. Almost 98% of the total weight of EVOO is represented by fatty acids, which constitute the saponifiable fraction of olive oil. The fatty acid content of EVOO consists of 75% monounsaturated fatty acids (mostly oleic acid), 11% polyunsaturated fatty acids (mostly linoleic acid), and 14% saturated fatty acids (mostly palmitic acid) [20,21]. The remaining 2% of the total weight of EVOO is represented by the unsaponifiable fraction. The stability and flavor of olive oil are given by the components of the unsaponifiable fraction.

The unsaponifiable fraction is divided into the non-polar, non-water-soluble, solvent-extractable fraction after saponification of the oil, which contains squalene and other triterpenes, sterols, tocopherol (mainly alpha-tocopherol, or vitamin E), and pigments, and the polar fraction, water-soluble, which contains phenolic compounds, or polyphenols.

Polyphenols make up 18–37% of the unsaponifiable fraction of EVOO; these are responsible for most of the health benefits associated with taking EVOO. It is a heterogeneous group of molecules with important properties that are both organoleptic and nutritional [21]. Extra virgin olive oil has an average concentration of phenolic compounds of about 230 mg/kg [61], with a concentration of polyphenols ranging from 50 to 800 mg/kg [62,63]. The absorption efficiency of olive oil polyphenols in humans has been evaluated around 55–66 mmol% [64]. Tyrosol and hydroxytyrosol are two of the most important phenols in olive oil. Hydroxytyrosol is present in olive oil in the form of ester with elenolic acid to form oleuropein; the absorption in humans is dose-dependent, related to the phenolic content of olive oil [65].

#### 2.6.4. Poly- and Monounsaturated Fatty Acids, Serum Cholesterol Levels and Cardiovascular Disease

A meta-analysis by Mensink et al. [66] showed that under isocaloric, metabolic ward conditions, when carbohydrates in the diet were replaced by fatty acids, HDL increased and triglycerides decreased, while LDL increased. In addition, if polyunsaturated fats replace saturated fats, then a more marked decrease in serum LDL and triglyceride levels was observed. Authors also showed that replacing saturated fatty acids with unsaturated fatty acids increased the ratio of HDL to LDL cholesterol, thus obtaining the most favorable lipoprotein risk profile for CHD. Substitution of saturated fatty acids with carbohydrates had no favorable effect on the CHD risk profile.

Subsequently, Maki et al. [67], in their randomized, double-blind, controlled-feeding trial, showed that CO reduced total cholesterol (TC), low-density lipoprotein cholesterol (LDL), very low-density lipoprotein cholesterol (VLDL), non-high-density lipoprotein cholesterol (non-HDL), and ApoB concentration to a greater extent compared with EVOO intake (CO compared with EVOO intake: TC = −0.37 vs. 0.02 mmol/L, *p* > 0.001; LDL = −0.36 vs. −0.08 mmol/L, *p* > 0.001; VLDL = −0.03 vs. 0.04 mmol/L, *p* > 0.001; non-HDL = −0.39 vs. −0.04 mmol/L, *p* > 0.001). ApoB, an indicator of circulating small and dense, and therefore highly atherogenic, LDL, was lowered largely by CO, compared to EVOO intake (−9.0 vs. −2.5 mg/dL, *p* > 0.001). HDL-C concentration did not differ significantly between CO vs. EVOO intake (0.02 vs. 0.05 mmol/L, *p* = 0.112), but ApoA1, which is the major protein component of HDL particles in plasma, increased more with EVOO compared with CO intake (4.6 vs. 0.7 mg/dL, *p* = 0.016).

The Nurses’ Health Study [68] prospectively investigated the association between different types of dietary fat intake and the risk of coronary heart disease in a 14-year follow-up in a cohort of 80,082 women, aged between 34 and 59 years of age, without a history of CHD, stroke, cancer, hypercholesterolemia, or diabetes. The authors demonstrated that a 5% increase in energy intake from saturated fat was associated with a 17% increase, although not statistically significant in the relative risk (RR) of CHD (RR = 1.17; 95% CI = 0.97–1.41; *p* = 0.10) compared to the equivalent energy intake from carbohydrates. The authors also demonstrated that for each 2% increase in energy intake from trans-unsaturated fats, a significant 93% increase in the risk of CHD was associated (RR = 1.93; 95% CI = 1.43–2.61; *p* = 0.001). Finally, the authors demonstrated that while for each 5% increase in energy intake from monounsaturated fats, there was a non-statistically significant 19% decrease in the risk of CHD (RR = 0.81; 95% CI = 0.65–1.00; *p* = 0.05); with each 5% increase in energy intake from polyunsaturated fats, there was a significant 38% reduction in the risk of CHD (RR = 0.62; 95% CI = 0.46–0.85; *p* < 0.003). The authors also showed that replacing 5% energy from saturated fat with unsaturated fat resulted in a 42% reduction in CHD risk (95% CI = 0.23–0.56; *p* < 0.001), while replacing 2% of energy from trans unsaturated fat with un-hydrogenated, unsaturated fats was associated with a 53% decrease of the risk of CHD (95% CI = 0.34–67; *p* < 0.001). The authors concluded by confirming that the replacement of saturated fats (SF) and trans-unsaturated fats in the diet with non-hydrogenated monounsaturated and polyunsaturated fats favorably alters the lipid profile, but that reducing overall fat intake has little effect. 

W. C. Willett [69] confirmed these data in a subsequent review by concluding that trans-unsaturated fatty acids in hydrogenated vegetable oils have obvious negative effects and should be eliminated. He also stated that a further reduction in CHD rates is possible if saturated fats are replaced by a combination of poly- and monounsaturated fats and the benefits of polyunsaturated fats appear stronger. 

A subsequent pooled analysis of 11 cohort studies by Jakobsen et al. [70] confirmed the effects of the polyunsaturated fatty acids. The authors showed a significant association between PUFA replacement and reduced risk of coronary events (HR: 0.87; 95% CI: 0.77, 0.97) and a significant association between PUFA replacement and reduced risk of mortality for CHD (HR: 0.74; 95% CI: 0.61, 0.89). In conclusion, the authors stated that, rather than increasing the consumption of MUFA or carbohydrates, increasing the consumption of PUFA in place of saturated fatty acids (SFA) could significantly prevent coronary heart disease among middle-aged women and men and among the elderly. 

Lai et al. [71] investigated the associations between de novo lipogenesis (DNL)-related fatty acids (FA) with total mortality and specific cause mortality including cardiovascular disease (CVD), CHD, and stroke, analyzing the data from the Cardiovascular Health Study (CHS) [72], measured at three time points over 13 years. Surprisingly, they found a direct association between higher oleic acid levels (18: 1n-9) and a high risk (hazard risk, HR) of all-cause mortality (HR = 1.56, 95% CI = 1.35–1.80, *p* < 0.001) including CVD and non-CVD mortality (HR = 1.48, 95% CI = 1.21–1.82, *p* < 0.001; HR = 1.50, CI = 95% 1.28–1.75, *p* < 0.001, respectively). They also found an association between higher oleic acid levels and fatal and non-fatal CVD, fatal and non-fatal CHD, fatal and non-fatal stroke (HR = 1.33, 95% CI = 1.12–1.57, *p* < 0.001; HR = 1.23, 95% CI = 1.01–1.48, *p* = 0.008; HR = 1.34, 95% CI = 1.02–1.75, *p* = 0.005, respectively).

Results of a meta-analysis by Borges et al. [73], which included five cohort studies and one matched case-control study, involving 23,518 subjects, showed that the risk (odds ratio, OR) of CHD was lower with higher circulating docosahexaenoic acid (DHA) levels (OR = 0.85; 95% CI = 0.76–0.95), but was not associated with stroke risk (OR = 0.95; 95% CI = 0.89–1.02); risk of stroke was lower with higher circulating linoleic acid (LA) levels (OR = 0.82; 95% CI = 0.75–0.90), but was not associated with CHD (OR = 1.01; 95% CI = 0.87–1.18); circulating MUFA were associated with higher CHD risk of stroke (OR = 1.22; 95% CI = 1.03–1.44) and CHD (OR = 1.36; 95% CI = 1.15–1.61). SFA was not related both with increased CHD risk (OR = 0.94; 95% CI = 0.82–1.09) and with stroke risk (OR = 0.94; 95% CI = 0.79–1.11).

Finally, Lee et al. [74] studied the associations between plasma AF levels with the risk of incident heart failure (HF) by analyzing data from CHS. They showed that plasma habitual levels and changes in the levels of palmitic acid (16:0) were associated with higher risk of HF (HR = 1.17, 95% CI 1.00–1.36; HR = 1.26 95% CI 1.03–1.55, respectively); plasma habitual levels of 7-hexadecenoic acid (16:1n-9) were not associated with risk of HF (HR = 1.05, 95% CI 0.92–1.18), but changes in levels were associated with a higher risk of HF (HR = 1.36, 95% CI 1.13–1.62); plasma habitual levels of vaccenic acid (18:1n-7) were not associated with risk of HF (HR = 1.06, 95% CI 0.92–1.22), but changes in levels were associated with a higher risk of HF (HR = 1.43, 95% CI 1.18–1.72); habitual levels and changes in levels of myristic acid (14:0) (HR = 0.90, 95% CI = 0.77–1.05; HR = 1.11, 95% CI = 0.91–1.36, respectively), palmitoleic acid (16:1n-7) (HR = 1.01, 95% CI = 0.88–1.16; HR = 1.06, 95% CI = 0.87–1.28, respectively), stearic acid (18:0) (HR = 0.94, 95% CI = 0.81–1.09; HR = 0.94, 95% CI = 0.76–1.15, respectively), and oleic acid (18:1n-9) (HR = 1.13, 95% CI = 0.98–1.30; HR = 1.13, 95% CI = 0.93–1.37, respectively) were not associated with HF risk, 

Despite these conflicting results, the advice to replace saturated fats with polyunsaturated fats in the diet remains a cornerstone of international guidelines for reducing the risk of CHD.

On the other hand, it would be overly simplistic to say that replacing SFAs with MUFA (oleic or linoleic acid) or PUFA may be sufficient in reducing the risk of CVD or mortality risk. The benefits of taking MUFAs are observed when they are associated with the concomitant intake of polyphenols and other natural antioxidants, contained, for example, in EVOO. In fact, there is no evidence to suggest that simply replacing SFA with MUFA reduces the risk of CVD or mortality. Similarly, the benefits of daily intake of PUFA-n3 are attributable to PUFAs, but above all where they are associated, similarly to MUFAs, with the intake of polyphenols or other natural antioxidants, and as part of a healthy diet such as the traditional Mediterranean diet. All the above-mentioned studies are shown in Table 2.

### 2.7. Barley

Barley (Hordeum vulgare) is a cereal grain that is grown in temperate climates worldwide. It is one of the oldest cultivated cereals, originally in the Fertile Crescent area of the Middle East and Egypt. Barley is used commonly as animal fodder. Concerning human nutrition, two types of barley are commonly found: hulled barley, which requires a long cooking time and preventive soaking, and pearl barley, which undergoes a refining process (similar to whitening rice) to remove the outermost part. This can be used without prior soaking and cooking time is shorter. Barley is used for the preparation of soups and stews and also for cooking barley bread. From the coarse ground, coarse semolina is obtained, suitable for typical North African dishes similar to couscous. Roasted in the oven at temperatures around 170–180 °C, and very finely ground until obtaining a powder similar to flour, and freeze-dried, it is used to quickly prepare drinks by adding hot water or milk or used as a substitute for coffee. Roasted fine flours are also obtained from the roasting of barley and are used in the preparation of sweets or pastries. Barley grains are commonly made into malt as a source of fermentable material for beer and distilled beverages, like whisky. 

#### 2.7.1. Nutrients

A 100 g of barley provides 352 calories. Barley is made up of about 28% carbohydrates, 57% dietary fiber, 2% fat, and 20% protein. Barley is also a good source of B vitamins and minerals including copper, iron, magnesium, manganese, phosphorus, selenium, and zinc [21].

#### 2.7.2. Barley β-Glucan

Β-glucan constitutes approximatively 75% of dry weight of endosperm cell walls, and arabinoxylan constitutes 25% [75]. The percentages of beta-glucan content in the barley grain vary according to the different polymorphisms of the genes that encode the corresponding synthase and endohydrolase enzymes [76]. Barley β-glucans also have cholesterol-lowering properties [77,78], however, lower than oats [79]. In addition, it is known that β-glucans from barley reduce post-prandial glycemic response with lowering blood glucose. This effect is due not because of the high viscosity of the β-glucans, but rather to the direct inhibition of the activities of glucose transporters and intestinal brush border enzymes [80,81]. More evidence has shown that β-glucans exert their beneficial effects on lipid and glucose metabolism and reduce CVD risk by the increase in colonic microbial population and activity, particularly favoring the increase in Lactobacillus over Bacteroidetes spp, yielding short-chain fatty acids as end products [48,82]. In animal models, these health benefits, on microbial gut flora, are also associated with an increase in lifespan and better locomotor activity, muscle coordination, and balancing activity [83]. 

## 3. Diet Pattern and Risk of Frailty and Mortality

It is now known that fruits and vegetables, in addition to the intake of whole grains, monounsaturated and omega-3 fatty acids, and moderate amounts of alcohol are fundamental elements of a cardioprotective diet [84]. Furthermore, it is known that a prevalent consumption of fruit, vegetables, and WG, even in a dietary model different from traditional MD, is protective (i.e., associated with a reduced risk of frailty) [85].

Lo et al. [86] analyzed data from the Nutrition and Health Survey in Taiwan. They showed that elderly subjects in the higher tertile of the dietary pattern score (i.e., with a high consumption of fruit, nuts and seeds, tea, vegetables, WG, omega-3-rich deep-sea fish, and shellfish and milk as protein-rich foods) had a reduced risk (Odds Ratio, OR) of frailty (OR = 0.12, 95% CI 0.02–0.76, *p* = 0.019) or pre-frailty (OR = 0.40, 95% CI 0.19–0.83, *p* = 0.015).

Following the studies by Ancel Keys, MD has been proposed as a model of healthy eating, associated with a reduced risk of developing cardiovascular and metabolic diseases [5]. Subsequently, Trichopoulou et al. showed that high adhesion to MD was associated with a reduction in the risk of total mortality [6,87].

Subsequently, the PREDIMED study [8,9,88] demonstrated that high-risk CVD subjects who followed an MD pattern, in which monounsaturated and antioxidant fatty acids came from taking EVOO, or alternatively taking omega-3 fatty acids from nut consumption, had a reduced risk of acute myocardial infarction, stroke, or death from CVD (MD with EVOO: HR = 0.70, 95% CI: 0.53–0.91, *p* = 0.009; MD with nuts: HR = 0.70, 95% CI: 0.53–0.94, *p* = 0.02), but not of total mortality (MD with EVOO: HR = 0.81, 95% CI: 0.63–1.05, *p* = 0.11; MD with nuts: HR = 0.95, 95% CI: 0.73–1.23, *p* = 0.68).

A meta-analysis [10] that involved 1,574,299 subjects followed for a time period of 3–18 years found a significant direct association between higher adherence to MD, improved health status, and reduced mortality risk (Rate Risk, RR) (RR = 0.91, 95% CI 0.89–0.94; *p* < 0.0001), particularly in mortality due to CHD (RR = 0.91, 95% CI: 0.87–0.95, *p* < 0.0001) and cancer (RR = 0.94; 95% CI: 0.92–0.96; *p* < 0.0001).

Another meta-analysis by the same authors [11] also showed a significant association between higher MD adherence, improved health and quality of life, and reduced overall mortality (RR = 0.92, CI 95%: 0.90–0.94, *p* < 0.00001). In particular, the authors showed a significant reduction in mortality from CHD (RR = 0.90; 95% CI: 0.87–0.93; *p* < 0.00001) or from cancer (RR = 0.94; 95% CI: 0.92–0.96; *p* < 0.00001).

Kromhout et al. [89] confirmed the association between a higher adherence to a dietary model with the characteristics of MD with a reduction in CHD mortality (r = −0.91). The authors further highlighted the protective role in the diet of cereals (r = −0.52), vegetables (r = −0.52), and legumes (r = −0.62) as well as the intake of a moderate amount of alcohol (r = −0.54).

Subsequently, Zaslavsky et al. [90] analyzed a sample of 10,431 women aged 65–84 years from the Women’s Health Initiative Observational Study [91,92] with complete frailty according to Fried’s criteria [93]. MD pattern adherence was assessed using the alternative MD (aMed) index [6,94], which considered the intake of fruit, vegetables, nuts, legumes, WG, fish, ratio of monounsaturated to saturated fat, red and processed meats, and alcohol. The authors further showed the association between a higher intake of vegetables, nuts, and WG with a significant reduction in mortality risk (HR = 0.91, 95% CI: 0.84–0.99, *p* = 0.02; HR = 0.87, 95% CI: 0.80–0.94, *p* < 0.001; HR = 0.83, 95% CI: 0.77–0.90, *p* < 0.001, respectively). The relative contribution of these components to the reduction of mortality risk, obtained by subtracting each component from the aMed Index, was respectively 21% (vegetables), 42% (nuts) and 57% (WG).

More recently, Campanella et al. [95] performed a survival analysis involving 4896 subjects from Castellana Grotte and Putignano (Apulia, Italy) included in the MICOL study [96] and in the NUTRIHEP study [97], respectively. The relative Mediterranean scoring system (rMED) [98] was used to measure adherence to MD. The rMED considers the intake of fruit (excluding fruit juices), vegetables (excluding potatoes), legumes, cereals, fresh fish, olive oil, meat and dairy products, and alcohol. The authors noted that higher MD adherence was directly correlated with longer lifespan. In particular, among subjects with greater adherence to MD at the baseline, the mean time to death was estimated to be postponed from 6.21 to 8.28 years compared to subjects with lower MD adherence.

The protective effect of the MD [99] are certainly due to the lipid-lowering effect, protection against oxidative stress, inflammation, and platelet aggregation, modification of hormones and growth factors involved in the pathogenesis of cancer, inhibition of nutrient sensing pathways by specific amino acid restriction, and gut microbiota-mediated production of metabolites influencing metabolic health. Specifically, the moderate energy restriction provided by the high consumption of fiber-rich energy-poor plant foods and the specific restriction of sulfur compounds, branch-chain amino acids, and saturated fatty acids, characteristics of the MD, play a prominent role in mediating the beneficial effects on the health and longevity of this dietary model. In addition, the intestinal microbiome, which is actively involved in the processing of many plant foods rich in fiber as well as several vitamins and phytochemicals, plays a vital role in maintaining both metabolic and molecular health.

Hernaez et al. [100] reported the results of a study conducted on a subsample of 296 subjects at high cardiovascular risk, extracted from the cohort of the PREDIMED study [8,9]. The authors confirmed the beneficial effects of the intake of EVOO, nuts, legumes, WG, and fish. Mostly, they showed that increase for one year in the intake of these cardioprotective foods was linked to an improvement in HDL biological functions. In particular, increase in daily intake of 10 g of EVOO and 25 g of whole grain was associated with increment in cholesterol efflux capacity, in other words, the capacity of HDL to pick up cholesterol (+0.7%, *p* = 0.026; +0.6%, *p* = 0.017, respectively). Increase in daily intake of 30 g of nuts and 25 g of legumes and 25 g of fresh fish was linked to increment in the activity of paraoxonase-1, a key HDL-bound antioxidant enzyme (+12.2%, *p* = 0.049; +11.7%, *p* = 0.043; +3.9%, *p* = 0.030, respectively). Increase in legumes and fish consumption was also related to decreases in the activity of cholesteryl ester transfer protein, pro-atherogenic when excessively active (–4.8%, *p* = 0.028; –1.6%, *p* = 0.021, respectively). 

The above evidence reaffirms a fundamental concept, namely, that it is not the single nutrient or the single antioxidant that is effective in reducing mortality, nor the risk of frailty, but the set of nutrients in the diet. Another key point is that diet is not intended as an effective therapy or as something to be taken for a defined time. In contrast, the diet should be understood as a diet to be practiced for life and in the context of a healthy lifestyle, as observed, for example, in populations following the traditional MD or the Okinawa diet.

All the above-mentioned studies are shown in Table 3.

## 4. Whole Grains Intake, Cardiovascular Risk Factors, and Body Weight 

According to the HEALTHGRAIN Consortium definition [101], whole grain (WG) means “the intact, ground, cracked or flaked kernel after the removal of inedible parts such as the hull and husk. The principal anatomical components, as the starchy endosperm, germ, and bran, are present in the same relative proportions, as they exist in the intact kernel. Small losses of components, that is, less than 2% of the grain/10% of the bran, that occur through processing methods consistent with safety and quality are allowed”.

Maras et al. [102] analyzed data from the Baltimore Longitudinal Study on Aging [103] and identified the main sources of WG as breakfast cereals (57.5%), multi-grain and whole wheat bread (16.5%), corn chips snack type (4.2%), popcorn (3.8%), and rye bread (3.6%).

Sette et al. [104] calculated whole grain intakes in an Italian sample of 2830 adults and older adults and of 440 children and adolescents from the INRAN-SCAI 2005–06 Study. The main source of total WG intake among adults and older adults were bread (46%), biscuits (20%), savory fine bakery products (15%), breakfast cereals (7%), and wheat and other cereals (6%). 

Subsequently, Ruggiero et al. [105] calculated WG intakes in a different Italian sample of 2830 adults and older adults and of 440 children and adolescents from the Italian Nutrition & Health Survey (INHES) Study. In this study, the major food sources of WG among adults and older adults were bread (53.3%), biscuits (27.4%), pasta (13.1%), breakfast cereals (4.8%), and soups (1.3%). Figure 4 summarizes the different intake of whole grain among U.S. and Italian populations. 

There is now growing epidemiological evidence that WG exerts beneficial effects on human health, especially concerning the metabolic profile [106]. In particular, the consumption of WG has been associated with a reduction in cardiovascular risk factors such as postprandial insulin, blood lipid profile, and finally, the intestinal microbiome [107,108,109], as summarized in Figure 5.

Kelly et al. [110] conducted a systematic review to evaluate the effect of WG diets on total cardiovascular mortality, cardiovascular events, and cardiovascular risk factors (blood lipids, blood pressure). Nine randomized clinical trials (RCTs) published from 2008 to 2014 were included involving 1414 subjects. All included studies reported the effect of WG on major CVD risk factors such as body weight, blood lipids, and blood pressure. The authors did not find any study that clearly reported any effect of WG diets on total cardiovascular mortality or on cardiovascular events (i.e., total myocardial infarction, unstable angina, coronary artery bypass graft surgery, percutaneous transluminal coronary angioplasty, total stroke). Furthermore, the authors specified that all studies involved primary prevention populations and had an unclear or high risk of bias, and no studies had a duration of intervention greater than 16 weeks. 

Kirwan et al. [111] reported the results of a double-blind randomized case-control study conducted in a sample of 40 men and women aged <50 years, with no known history of CVD but who were overweight or frankly obese to compare the effects on body composition and metabolism of a diet containing WG versus an energy diet with refined grains. Each group followed the two diets for eight weeks; a washout period of 10 weeks was interposed between the two diets. The authors described an improvement in diastolic blood pressure (DBP) among overweight and obese adults that was >3 times greater at the end of the feeding period with the WG diet compared to the period of consumption of refined grains (−5.8 mm Hg, 95% CI: −7.7–−4.0 mm Hg; −1.6 mm Hg, 95% CI: −4.4–1.3 mm Hg; *p* = 0.01, respectively). Regarding systolic blood pressure (SBP), the authors did not observe any significant differences in the magnitude of reduction between WG diet and refined-grain diet group (*p* = 0.80). In addition, the authors observed a lower decrease in plasma adiponectin levels after the whole-grain diet compared with the control diet (−0.1 μg/mL, 95% CI: −0.9–0.7; −1.4 μg/mL, 95% CI: −2.6–−0.3, *p* = 0.05, respectively). The preserved total circulating adiponectin concentrations were related to the concentration of circulating adiponectin (r = 0.35, *p* = 0.04).

Subsequently, Marventano et al. [112] performed a meta-analysis including 41 RCTs to evaluate the effect of WG-containing foods on glycemic control and insulin sensitivity in healthy individuals in the short-, medium-, and long-term by analyzing changes from baseline fasting blood glucose and insulin levels and insulin levels by measuring the area under the curve (iAUC). The authors showed that WG foods induced a significant reduction in the post-prandial values of the glucose iAUC and of insulin iAUC at 120 min by −29.71 mmol min/L and by −2.01 nmol min/L, respectively. They concluded by stating that in healthy subjects, the consumption of WG foods improved postprandial glycemia, insulin response as well as insulin and glucose homeostasis compared to the consumption of refined grain derivatives. The authors suggested, as a possible mechanism that could explain these effects of WG, both the slower digestion rate and the action produced by the microbiome in the large intestine through the fermentation of resistant fibers and starches, with the consequent production of short-chain fatty acids (SCFAs). These short-chain fatty acids, once in the liver, would improve glucose homeostasis and insulin sensitivity by increasing glucose oxidation, the reduction of fatty acid release, and by augmenting insulin clearance [113].

In addition, Musa-Veloso et al. [114] conducted a meta-analysis on 20 full-text articles with the aim of evaluating the effects induced by the consumption of WG wheat, WG rice, or WG rye on postprandial glycemia by comparing the glycemic values after the consumption of the same refined grains. They reported that a significant reduction in blood glucose AUC was observed only after consumption of WG rice compared to white rice (−40.5 mmol/L x min; 95% C =−59.6–−21, 3; *p* < 0.001). In contrast, no significant change in blood glucose AUC was reported, either after the consumption of whole wheat, compared to white wheat, or after the consumption of whole-meal rye compared to refined rye (−6.7 mmol/L x min, 95% CI = −25.1–11.7, *p* = 0.477; −5.5 mmol/L x min; 95% CI = −24.8–13.8; *p* = 0.576, respectively).

Kirø et al. [115] analyzed data from the Diet, Cancer, and Health cohort study [116]. They reported a reduction in the risk of type 2 diabetes of 11% for men and 7% for women, for each increase in consumption of WG (mainly rye) of 16 g/day (HR = 0.89, CI 95% = 0.87, 0.91; HR = 0.93, 95% CI = 0.91. 0.96, respectively). The highest quartile group of WG consumption had a reduction in the risk of type 2 diabetes of 34% for men and 22% for women (HR = 0.66, 95% CI: 0.60–0.72, *p* < 0.0001; HR = 0.78, 95% CI: 0.70–0.86, *p* < 0.0001, respectively). The authors also observed a reduced risk of 12% type 2 diabetes mellitus for men and 7% for women for every increase in the consumption of WG products (mainly rye bread) 50 g/day (HR = 0.88, 95% CI = 0.86–0.90; HR = 0.93, 95% CI = 0.90–0.96, respectively). In addition, a 37% reduction in the risk of type 2 diabetes mellitus for men and 20% for women was observed in the highest quartile group of consumption of WG products (HR = 0.63, CI 95%: 0.58–0.69, *p* < 0.0001; HR = 0.80, 95% CI: 0.72–0.88, *p* < 0.0001, respectively).

Maki et al. [117] performed a meta-regression analysis of cross-sectional data from 12 observational studies involving 136,834 subjects, and a meta-analysis of nine RCTs (WG versus controls) that involved 973 subjects to examine the relationship of WG intake with body weight; they also qualitatively reviewed six prospective cohort publications. The meta-regression analysis from cross-sectional studies indicated a significant inverse correlation between WG intake and body mass index (BMI) (r = −0.526, *p* = 0.0001). The review of the results of the qualitative analysis from the prospective cohort studies, with a follow-up period from five to 20 years, showed an inverse correlation between WG consumption and body weight change. Meta-analysis of RCTs, with a length from 12 to 16 weeks, did not show any significant difference in weight change (standardized mean difference = −0.049 Kg; 95% CI = −0.388–0.199; *p* = 0.698). 

The discordant results are easily explained by the short duration of some of the studies. Twelve or even 16 weeks is too short a period to observe a significant reduction in cardiovascular events and mortality. The most significant results were observed in long-term prospective studies. This further reinforces the key concept that the intake of WGs should not be compared to taking a drug therapy, which in any case has effects in the short- and medium-term. The intake of WG in the diet must be contextualized, and the above evidence confirms it as part of a healthy diet.

All studies are summarized in Table 4.

## 5. Whole Grains Intake and Reduction of Mortality

Accumulating evidence indicates that high intake of WG decreases the risks of mortality from all causes, CVD, and cancer in the general population. 

Ma et al. [118] conducted a meta-analysis of prospective cohort studies involving 843,749 subjects and 101,282 deaths to quantify the association between WG intake and all-cause mortality. They showed that high WG intake was associated with a reduction of 18% for all-cause mortality risk (RR = 0.82, 95% CI = 0.78–0.87). In addition, the authors reported a 7% reduction in the risk of mortality from all causes for each increment of 16 g/day of WG consumption (RR = 0.93, 95% CI = 0.89 to 0.97). 

A subsequent interesting meta-analysis by Zong et al. [119] involving 786,076 subjects with 97,867 total deaths confirmed the association between WG intake and reduction in mortality. The authors showed that a high WG intake was associated with a significative reduction in total mortality, CVD mortality, and cancer mortality (RR = 0.84, 95% CI = 0.80–0.88, *p* < 0.001; RR = 0.82, 95% CI = 0.79–0.85, *p* < 0.001; RR = 0.88, 95% CI = 0.83–0.94, *p* < 0.001, respectively). The authors also estimated that each serving/day increase in WG intake was associated to a reduction of 7% for total mortality (RR = 0.93, 95% CI = 0.92–0.94), 9% for CVD mortality (RR = 0.91, 95% CI = 0.90–0.93), and 5% for cancer mortality (RR = 0.95, 95% CI = 0.94–0.96).

The meta-analysis by Wei et al. [120] involving 816,599 subjects with 89,251 all-cause deaths, 23,280 CVD deaths, and 35,189 cancer deaths obtained similar results. The authors found that a high WG intake was associated with a signification reduction in risk (summary relative risk, SRR) for total mortality, CVD mortality, and cancer mortality (SRR = 0.87, 95% CI = 0.84–0.90; SRR = 0.81, 95% CI = 0.75–0.89; SRR = 0.89, 95% CI = 0.82–0.96, respectively). The dose-response analysis showed a reduction in overall mortality risk of 19% as well as a reduction of CVD mortality risk and cancer mortality risk of 26% and 9%, respectively (SRR = 0.81, 95% CI = 0.76–0.85; SRR = 0.74, 95% CI = 0.66–0.83; SRR = 0.91, 95% CI = 0.84–0.98), for every three servings/day increase in WG consumption. 

Furthermore, a meta-analysis conducted by Aune et al. [36] confirmed the association between WG intake and reduction of mortality. The meta-analysis was performed on 45 prospective studies involving 245,012 to 705,253 participants with 7068 cases of coronary heart disease, 2337 cases of stroke, 26,243 cases of cardiovascular disease, 34,346 deaths from cancer, and 100,726 all cause deaths. In their study, the authors found that a high WG intake was associated with a signification reduction in CHD (RR = 0.79, 95% CI = 0.73–0.86), stroke (RR = 0.87, 95% CI = 0.72–1.05), and CVD (RR = 0.84, 95% CI = 0.80–0.87). The authors also reported a reduction in the risk of CHD, stroke, and CVD respectively of 19% (RR = 0.81, 95% CI = 0.75–0.87), 12% (RR = 0.88, 95% CI = 0.75–1.03), and 22% (RR = 0.78, 95% CI = 0.73–0.85) for each increase of 90 g/day (three servings/day) of the consumption of WG. In addition, the authors estimated that a high WG intake was associated with a signification reduction in mortality risk for CHD (RR = 0.65, 95% CI = 0.52–0.83), stroke (RR = 0.85, 95% CI = 0.64–1.13), CVD (RR = 0.81, 95% CI = 0.75–0.87), cancer (RR = 0.89, 95% CI = 0.82–0.96), and mortality for all-cause (RR = 0.82, 95% CI = 0.77–0.88). In particular, a reduced risk of coronary heart disease, stroke, CVD, cancer and overall mortality was observed by 19% (RR = 0.81, 95% CI = 0.74–0.89), 14% (RR = 0.86, 95% CI = 0.74–0.99), 29% (RR = 0.71, 95% CI = 0.61–0.82), 15% (RR = 0, 85, 95% CI = 0.80–0.91), and 17% (RR = 0.83, 95% CI = 0.77–0.90), respectively, for each increase in consumption of 90 g/day (three portions/day) of WG.

Another meta-analysis performed by Benisi-Kohansal et al. [121] involving 2,282,603 participants from 20 prospective cohort studies further confirmed the association between WG intake and reduction in mortality. Authors found that higher consumption of WG was associated with a reduction in overall mortality (RR = 0.87; 95% CI = 0.84–0.91), CVD mortality (RR = 0.84; 95% CI = 0.78–0.89), and cancer mortality (RR = 0.94; 95% CI = 0.91, 0.98). The authors also estimated a reduction of overall mortality, CVD, and cancer mortality of 17% (SRR = 0.83; 95% CI = 0.79–0.88), 25% (SRR = 0, 75; 95% CI = 0.68–0.83) and 10% (SRR = 0.90; 95% CI = 0.83–0.98), respectively, for each additional three servings/day (90 g/day) of the consumption of WG.

A further confirmation of the beneficial effects of WG consumption on the reduction of overall mortality risk as well as CVD and cancer mortality risk was provided by Zhang et al. [122], which conducted a meta-analysis on 19 prospective cohort studies involving 1,041,692 subjects. The authors confirmed the relationship between a high intake of WG and the risk reduction of all-cause mortality (RR = 0.84; 95% CI = 0.81–0.88). The authors also confirmed that higher WG consumption was related with a reduction in mortality risk for both CVD (RR = 0.83; 95% CI = 0.79–0.86) and for cancer (RR = 0.94; 95% CI = 0.87–1.01). After performing the dose-response analysis, the author estimated that each serving/day intake of whole grain could reduce the overall mortality by 9% (RR = 0.91; 95% CI = 0.90–0.93), CVD mortality by 14% (RR = 0.86; 95% CI = 0.83–0.89), and cancer mortality by 3% (RR = 0.97; 95% CI = 0.95–0.99). 

This latest evidence on the reduction in total cardiovascular mortality and even of mortality from neoplastic disease obtained from long-term perspective studies, further confirms the key concept that the benefits of taking WG cannot be interpreted as the beneficial effect of an isolated nutrient, but must be contextualized as part of a healthy diet in a healthy lifestyle. All studies are reported in Table 5.

It is therefore evident that a high intake of WG, vegetables, fruits, nuts, and coffee is associated with a reduced risk of mortality whereas a high intake of red and processed meat is related to a higher mortality risk. High-quality diets such as MD are associated with a reduced risk of all-cause mortality [123].

Few studies relate the effects of a diet high in fibers on gastrointestinal function, glycemic or lipid metabolism, or on body weight [124]. Gopinath et al. [125] examined the relationship between total dietary carbohydrate intake, glycemic index (GI), glycemic load (GL), and fiber intake, with the state of successful aging [3,4] and with mortality risk, for a follow-up period of 10 years in a cohort of 1609 adults from The Blue Mountains Eye Study [126]. The authors showed that higher intake of total fiber, and particularly vegetable fibers and fruit fibers, was associated with greater odds of successful aging (OR = 1.79, 95% CI = 1.13–2.84; OR = 1.26, 95% CI = 0.83–1.91; OR = 1.81, 95% CI = 1.15–2.83, respectively). 

Nevertheless, there have been a small number of studies examining the effect of WG on outcomes other than cardio-metabolic function or gastro-enteric function, or glycemic or lipid metabolism, or body weight. For example, there are very few studies analyzing the effect of WG in the diet on aging.

In this regard, Foscolou et al. [127] conducted an interesting study on a sample of 3349 elderly subjects from the ATTICA study and from the MEDIS study [36,128], both aimed to assess the association between WG intake with the diet and successful aging, and evaluated with the successful aging index (SAI) [129]. By applying the linear regression models, the authors observed a significant association between low vs. high intake of WG and SAI (b ± SE = −0.278 ± 0.091, *p* = 0.002). They did not observe any significant association between low vs. moderate WG intake and SAI (b ± SE = 0.010 ± 0.083, *p* = 0.901), and between moderate vs. high WG intake and SAI (b ± SE = −0.178 ± 0.095, *p* = 0.062).

## 6. Reduction of Protein to Carbohydrates Ratio Influence Aging and Lifespan

There is a consensus among gerontology researchers that dietary interventions can slow down aging, that is, prevent or delay the onset of numerous age-related chronic diseases [130,131,132]. The study of the relationship between nutrition and healthy aging has increasingly become a subject of great interest. Caloric restriction (CR), avoiding malnutrition, is the most studied dietary intervention known to extend life in many organisms [133,134,135]. To date, CR has been the focus of most non-genetic nutritional interventions. CR has been shown to improve several markers of health [136,137]. Fasting is the most extreme of the CR interventions, which requires the complete elimination of nutrients. Indeed, one of the more evaluated forms of fasting in both rodent and human studies is intermittent fasting [IF]. IF reduces body weight, body fat, and particularly abdominal fat, plasma insulin concentrations in both men and women, and reduces blood pressure, improving insulin sensitivity and lipid profile [138,139,140]. 

As reported by de Cabo and Mattson [141], cells exposed to fasting produce an adaptive stress response that leads to an increased expression of the antioxidant defenses, DNA repair, and control of protein quality, mitochondrial biogenesis, and autophagy, and downregulation of inflammation. In particular, the cell in intermittent fasting regimen showed a better and stronger resistance to a wide range of potentially harmful insults that involve metabolic, oxidative, ionic, traumatic, and proteotoxic stress. The protective effects of intermittent fasting are mediated by the stimulation of autophagy and mitophagy and by the inhibition of the mTOR (mammal target of rapamycin) protein synthesis pathway [142]. These responses allow cells to remove damaged proteins and mitochondria from oxidation and to recycle the molecular constituents not damaged by temporarily reducing the overall protein synthesis to conserve energy and molecular resources. In humans, intermittent fasting interventions induce health benefits than can largely be attributed to simply reducing caloric intake. These benefits are due to the loss of fat mass, and consequently to the decrease in fasting insulin levels and to the increase in insulin sensitivity, resulting in a decrease of insulin resistance, dyslipidemia, hypertension, and a pro-inflammatory state typical of advanced age. Nevertheless, the IF needs medical supervision as it may cause serious adverse effects in patients with extremely low BMI or between the frail and elderly patients [132].

Further evidence has suggested that macronutrient balance, rather than simple calorie restriction, plays a more important role in extending lifespan [143,144,145]. That is, modulating protein and carbohydrate intake, rather than simplistically reducing the entire energy intake, may offer a more feasible nutritional intervention in humans [146]. That is, it has become clear that specific nutrients and nutritional balance (i.e., the result of interactions between nutrients) play an important role in the biology of aging. 

The “Geometric Framework for Nutrition” (GNF) [147,148] is a method developed in nutritional ecology with the purpose of understanding the nutritional interactions of animals with their environments by explicitly distinguishing the roles of calories, individual nutrients, and nutrient balance. In this model, the nutritional requirements of an animal can be schematized in a two-dimensional or three-dimensional Cartesian space, called the nutrient space. The axes that define this space each represent a functionally important food component, for example, proteins, carbohydrates, and fats. The intake target (IT) represents the balance and quantity of functional nutrients for regulatory mechanisms (e.g., proteins and carbohydrates). The animal can reach the IT if appropriate foods are available. As shown in Figure 6, foods are represented by radials or nutritional tracks (T), which are projected into the space between nutrients according to angles determined by the ratio of the nutrients they contain. The animal can reach its target state by selecting Food 1, which is nutritionally balanced with respect to its target, or by mixing its intake with nutritionally complementary foods (Food 2 and Food 3). Therefore, when on the T1, the animal is unbalanced toward Food 2, that is, off course with respect to its target; however, it can get closer to the target by approaching Food 3, then passing into T2, and a further passage to Food 2 brings it closer to its nutrition target.

In this regard, basic research studies conducted on Drosophila have shown that longevity was maximal when the diet included a 1:16 ratio of proteins to carbohydrates, while reproductive capacity, measured in insects through egg production, was maximum with a ratio of proteins and carbohydrates between 1:2 and 1:4. This increased lifespan observed with a low-protein diet was attributed to a reduction in initial mortality and a delayed acceleration of age-dependent mortality [149,150]. Further studies conducted on male decorated crickets have confirmed that a low protein and high carbohydrate diet could induce higher immune functions and consequently lower mortality [151]. Similar results were observed in studies conducted on Gasterosteus aculeatus, or stickleback fish, where a significant increase in lifespan was observed in fish with a diet with a lower protein content than carbohydrates, as opposed to an increased reproductive capacity observed in a diet higher in protein than carbohydrates [152]. 

Evidence showed that these models of a low protein, high carbohydrate diet would induce a reduced TOR signaling [145,153]. In this regard, Senior et al. [154] examined data from the Solon-Biet study [145] to evaluate how the macronutrient content in the diet could influence life expectancy and mortality in a sample of mice. They showed that the mouse’s self-selected diet, which was composed of 22% protein, 47% carbohydrate, and 31% fat, with a protein–carbohydrate ratio lower than one, was associated with a long-life expectancy, low mortality in early and middle age, and high mortality in old age. In contrast, a diet rich in proteins or fats relative to carbohydrates produces low life expectancy with high mortality rates across all age classes. 

There are no studies on humans that have applied nutritional geometry. However, the great longevity of populations from Sardinia in Italy or from Okinawa in Japan is well known as is the generally low mortality of the populations of the Mediterranean basin who follow a traditional Mediterranean diet [6,7,16]. The explanation for this high life expectancy lies in traditional eating habits, both in the Mediterranean diet and in the Okinawan diet, which are rich in carbohydrates taken with cereals (wheat or rice) or its derivatives and with low protein content, with a protein/carbohydrate ratio for both diets of about 1:10 [155].

Regarding protein intake, Pedersen et al. [156] conducted an interesting review that aimed to assess the health effects of protein intake in healthy adults. The 64 papers that were included in the study were classified according to the grade of evidence as “convincing”, “probable”, “suggestive”, or “inconclusive”. The authors assessed as “suggestive” the evidence regarding the increased risk of all-cause mortality in relation to a low carbohydrates high protein (LCHP) diet, where total protein intake of at least 20–23% of total energy; they also assessed as “suggestive” the evidence concerning relations between vegetable protein intake and low risk of cardiovascular mortality. 

With regard to the carbohydrates, it is known that molecules are essential for many cellular processes, mainly for the production of energy, after having been converted into glucose by the cells. Furthermore, high blood glucose levels are known to be responsible for the progression of chronic diseases such as diabetes mellitus. More importantly, glucose is one of the most studied nutrient molecules, influencing lifespan in various model organisms. For example, diets enriched with glucose reduce lifespan in Caenorhabditis elegans by inhibiting the insulin/IGF-1 (IIS) signaling pathway. Diets enriched with glucose act by inhibiting DAF-16/FOXO, HSF-1, and SKN-1/nuclear factor erythroid-related factor (NRF), which regulate the expression of several target genes in the IIS pathway. Treatments with high glucose content also produce negative effects of aging on human endothelial progenitor cells (EPCs) and fibroblasts. In these human cells, elevated glucose treatment accelerates various aging-related phenotypes through the activation of the p38 mitogen-activated protein kinase (MAPK). High-glucose treatments induce downregulation of sirtuins; this leads to a reduction in FOXO activity and accelerates cellular senescence. In these glucose-rich conditions, we observed in the EPC some cellular aging phenotypes such as increased levels of b-gal staining SA, reduced cell proliferation, irregular morphology, and increased levels of ROS [150].

Regarding the relation between dietary carbohydrate intake and mortality, Seidelmann [157] conducted a prospective cohort study aimed at investigating the association of carbohydrate intake with mortality and residual life span in a large cohort of adults from the risk of community atherosclerosis (ARIC), which involved 15,428 subjects with a 25-year follow-up. The authors also investigated whether the replacement of carbohydrates with animal or vegetable sources of fats and proteins changed the observed associations. As a benefit of the study, they combined their findings with data from seven studies from North America, Europe, Asia, and multinationals involving 432,179 participants to contextualize all findings in a meta-analysis. The author showed that an increased risk of mortality was related to low carbohydrate consumption (low versus moderate carbohydrate consumption: HR = 1.20; 95% CI = 1.09–1.32; *p* < 0.0001) and high carbohydrate consumption (high versus moderate carbohydrate consumption: HR = 1.23; 95% CI = 1.11–1.36; *p* < 0.0001). After exploring the association between mortality and alternative source of fat and protein to carbohydrate intake, the authors found that increasing the protein and animal fat intake instead of carbohydrates was associated with a significantly increased mortality risk (HR = 1.18; 95% CI = 1.08–1.29; *p* < 0.0001). Alternatively, an increased intake of protein and vegetable fats instead of carbohydrates has been related to a significant reduction in mortality risk (HR = 0.82; 95% CI = 1.78–1.87; *p* < 0.0001). In conclusion, the authors stated that a diet with a low carbohydrate content, in which carbohydrates are replaced with fat and protein mainly of plant origin, might be associated with a higher life expectancy and a long-term approach could be considered to promote healthy aging. It would be simplistic, in fact, if we wanted to evaluate the effects of nutrition on aging and longevity, consider separately the intake of proteins or carbohydrates, or extrapolate from the context caloric intake total daily, for example, by simply reducing energy intake.

## 7. Discussion 

The application of the GNF method to study the effects of macronutrients and caloric intake on aging and lifespan has allowed us to understand that not the single nutrient but the interaction between macronutrients affects age-related health and lifespan. Animal model studies, which applied the GNF method, have shown that a low-protein, high-carbohydrate diet increases lifespan in many species. These studies were performed using ad libitum feeding regimes, which considered the eating behavior in a not strictly organized environment (i.e., less artificial), which produces more reliable results, if we want to translate them into human populations. Food intake is recorded so that diet content and calorie and macronutrient intake can be assessed [158]. All these studies have shown that longer lifespans are generated by diets low in protein and high in carbohydrates (low protein high carbohydrates, LPHC diet) where the optimal ratio of protein and carbohydrates is about 1:10, with the protein content of diet about 10% or less. All these studies have also shown that simple reduced caloric intake has no effect or negative effect on lifespan, which is clearly in contrast to many previous studies on CR.

Although CR overall has shown established benefits for health and aging, this model is not readily feasible in practice in both humans and animals that have free access to food. In contrast, alternative dietary models such as the LPHC diet, which allow ad libitum access to food, are more likely to be feasible as health interventions. Comparing the results of LPHC diets with those of caloric restriction, we can find similarities such as the reduction of insulin and inactivation of mTOR, and differences, the most interesting of those concerning mitochondrial biogenesis between cellular mechanisms concerning aging and life span [159]. Interestingly, LPHC diets are associated with reduced mitochondrial number and reduced expression of the master regulator of mitochondrial biogenesis, peroxisome proliferator-activated receptor gamma coactivator 1-alpha [PGC-1α], unlike CR, where there is an increase in the number of mitochondria associated with a greater expression of PGC-1α.

The concept of “mythormesis” could explain the paradox that both LPHC and CR diets increase lifespan but induce opposite effects on mitochondria. After postulating that low levels of oxidative stress induce the activation of systemic defense mechanisms beneficial for aging such as the activation of endogenous antioxidant enzymes, LPHC diets can increase the production of hydrogen peroxide sufficiently to generate hormetic benefits without producing mitochondrial damage.

It is now an irrefutable fact that a healthy lifestyle during a younger age, which includes consuming healthy foods in an amount appropriate to both health and physical activity, smoking cessation, and even taking moderate amounts of alcohol, prepares for successful aging. It is also an irrefutable fact that one of the main factors in increasing the average life span in the last two centuries has been the improvement in the nutritional status of the population. In contrast, a poor-quality diet is still the main risk factor of mortality, but above all, of disability in older age, even in developed and wealthy nations [160]. 

The most recent evidence has shown that diets that are rich in low glycemic index carbohydrates, combined with low amounts of proteins, are optimal to determine a longer and healthier life expectancy. In addition, diets that combine high amounts of refined, starch rich, high glycemic index carbohydrates with high contents of animal-derived proteins and fats determine a higher mortality rate, especially from CVD [161,162,163,164].

Dietary patterns that demonstrated greater adherence to diets that emphasized fruit and vegetables, WG rather than refined grains, low-fat dairy, lean meats, legumes, and nuts were inversely associated with mortality [165,166]. We already know that adherence to a diet pattern such as the traditional MD [6] was associated with a reduction in overall mortality, coronary heart disease, and cardiovascular disease. An overall high-quality diet that emphasizes a high consumption of polyunsaturated and monounsaturated fatty acids, raw vegetables, dairy, legumes, low-fat lean meats fat, fresh fish, bread (especially whole-grain bread), and wine in moderation, has been inversely associated with overall mortality, especially in the elderly [167,168]. In addition, a healthy diet of high quality can increase the number of years without disease and without disabilities [169]. Evidence also suggests that higher adherence to a dietary pattern that includes mainly legumes, fruit, vegetables, cereals, bread, olive oil, and dairy products, more occasionally meat, fish, and seafood, is associated with lower risk of becoming frail in old age [170]. In contrast, an eating style characterized by a high consumption of refined cereals has been associated with a greater risk of total mortality, especially mortality from major CVD [171].

Concerning protein intake, an established dietary bias is that older people need to obtain more protein with their diet, even though they are not malnourished. The main objective of this advice is to first prevent sarcopenia, then to maintain a good state of health, which allows for the prevention of malnutrition, improving wound healing and faster recovery from acute illness [167]. Instead, these recommendations are in contrast with the results of basic research on animal models and with the results of observational studies on population cohorts, which on the contrary have shown that a low-protein, high-carbohydrate diet (LPHC) can delay aging and extend lifespan [123,133,142,159,172]. 

Residents of the Japanese island of Okinawa and people living in the central-eastern mountainous area of the Italian island of Sardinia [173,174], although so distant, share a unique characteristic: both populations show one of the highest concentrations of centenarians in the world, whose ages have been carefully validated. Several factors contribute to the exceptional longevity of these populations. Among these are a moderate caloric intake that is never excessive, a high quality of food, constant physical activity, and genetic predisposition.

Regarding centenarians in Okinawa, their dietary energy intake comes from 85% carbohydrates and only 9% from proteins. [175]. Furthermore, the ratio of proteins to carbohydrates is extremely low [1:10], like what has been discovered to optimize lifespan in aging studies in animal models [159].

Concerning Sardinian centenarians [174], the consumption of sourdough bread, which is prepared from WG with a microbial yeast containing lactobacilli called “mother yeast”, with chemical and physical characteristics quite different from bread bought from the ovens, and a plant soup called “minestrone” containing fresh vegetables (onions, fennel, carrots, celery) and legumes (beans, broad beans, peas) is very widespread. Honey was generally used as a sweetener. Meat consumption does not exceed 2–4 portions per month. As for the consumption of dairy products, these people make extensive use of ricotta (whey cheese and dry curd), both goat and sheep, rather than mature cheese, and a local fresh sour cheese called “casu axedu” in the local dialect, which is rich in lactobacilli.

Indeed, the consumption of sourdough bread is very widespread in the traditional MD in southern Italy. This type of bread can reduce blood glucose and postprandial insulin levels by 25%, thus being able to preserve the function of pancreatic insulin-secreting cells and prevent obesity and diabetes [176].

Other predisposing health promoting factors between both people from Sardinia and from Okinawa are physical activity, low stress levels, and strong community support. We have already stated [177] that the benefits of MD should not be simply attributed to the high content of fiber, antioxidants, and proteins of vegetable origin. Nevertheless, it should be reiterated that the benefits of MD should be considered as part of a cultural context where food, together with the convivial aspect, is part of a “Mediterranean” lifestyle [178,179].

## 8. Conclusions

Longevity is the result of a multifactorial phenomenon that also involves nutrition. Among the main causes of the increase in lifespan in the last two centuries, we certainly recognize that the improvement in the nutritional status [180,181,182], while paradoxically an energy-intensive diet but of low nutritional quality widespread in the last century in developed countries, represents the main risk factor for mortality and disability [160]. Many studies have shown that a higher consumption of proteins and fats is related to a reduction in life expectancy, while high intake of low glycemic index carbohydrates from WG might play a protective role. Indeed, evidence shows that regular WG intake reduces the risks of cardiovascular disease and stroke, hypertension, metabolic syndrome, and diabetes as well as several forms of cancer [183]. Furthermore, more recent evidence shows that, instead of simply reducing overall calorie intake, ad libitum access to foods as part of a low-protein, high-carbohydrate diet extends lifespan. In conclusion, a high healthy life expectancy is the result of several factors. The most important undoubtedly include a healthy lifestyle with continuous physical activity, abstention from smoking, and intake of moderate quantities of alcohol, combined with a healthy diet in close symbiosis with lifestyle. In particular, optimizing caloric intake in relation to physical activity and age-related changes in metabolism, regular intake of whole-grain derivatives, together with the optimization of the protein/carbohydrate ratio in the diet, where the ratio is significantly lower than 1 such as in the traditional MD and the Okinawa diet, increases healthy life expectancy by reducing the risk of developing CVD and aging-related diseases.

## Figures and Tables

**Figure 1 nutrients-13-02540-f001:**
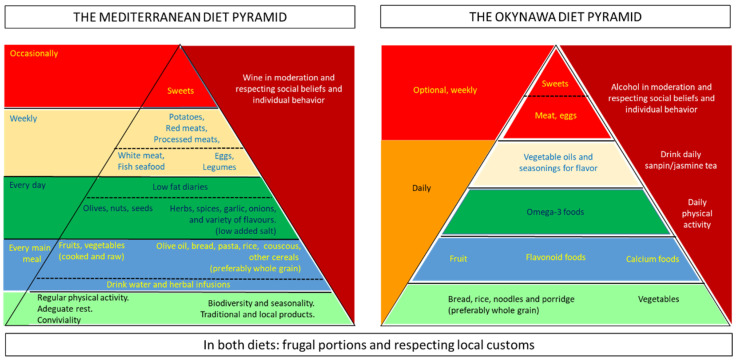
Mediterranean diet and Okinawan diet pyramids.

**Figure 2 nutrients-13-02540-f002:**
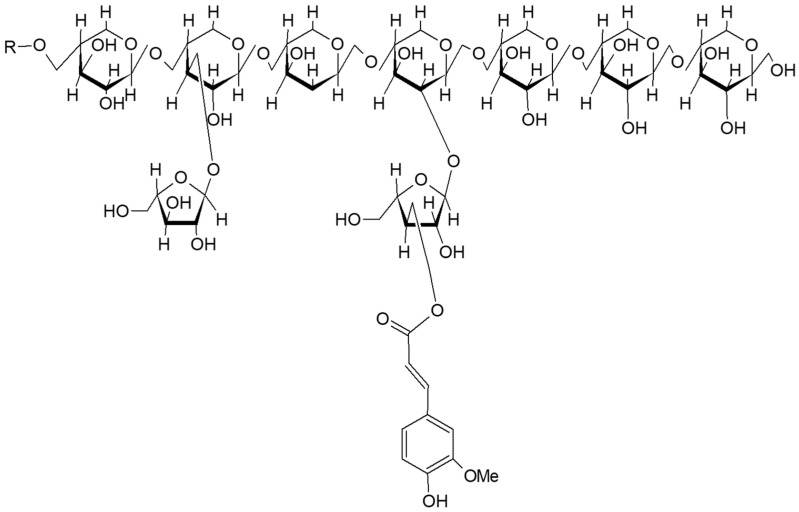
Arabinoxylan (drawn by ACD/ChemSketch).

**Figure 3 nutrients-13-02540-f003:**
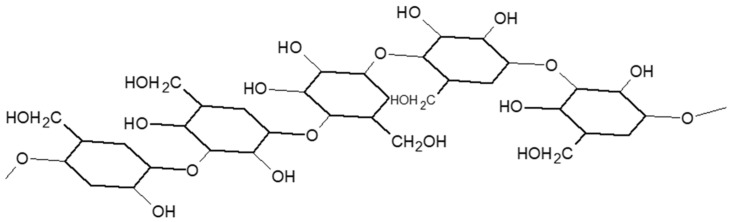
Beta-D-glucan (drawn by ACD/ChemSketch).

**Figure 4 nutrients-13-02540-f004:**
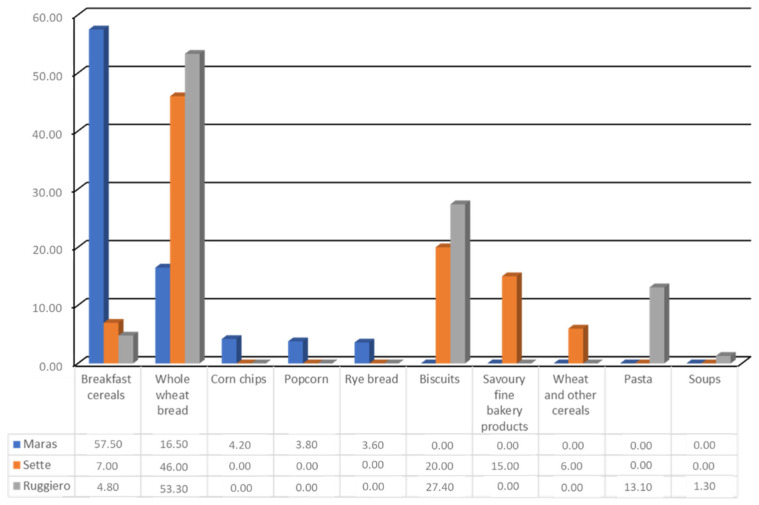
Intake of whole grain: U.S. vs. Italian population. Adapted from Ruggiero et al. [105].

**Figure 5 nutrients-13-02540-f005:**
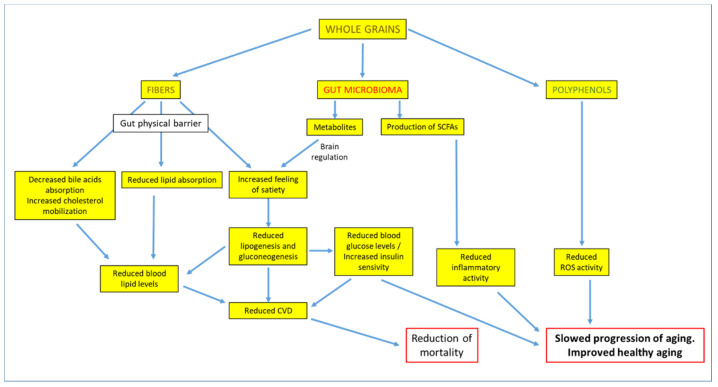
WG(whole grain)’s main mechanisms in reducing mortality and slowing aging.

**Figure 6 nutrients-13-02540-f006:**
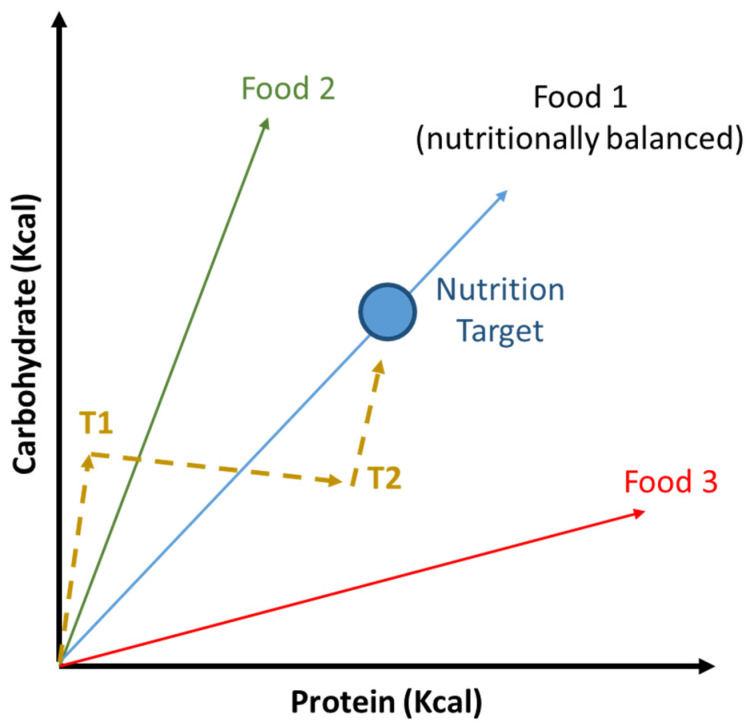
Dietary imbalance in nutritional geometry. Adapted from Raubenheimer et al. [148].

**Table 1 nutrients-13-02540-t001:** Nutritional properties of cereals.

	Wheat (Variety Hard, Red Winter)	Spelt (Uncooked)	Oats	Rye	Barley (Raw and Pearled)	Rice (Unenriched White Rice)	Maize (Sweetcorn, Yellow, Raw)
**Energy (KJ)**	1368	1415	1628	1414	1473	1498	1506
**Protein (g)**	12.61	14.57	16.89	10.34	9.91	6.5	3.27
**Global fats (g)**	1.54	2.43	6.9	1.63	1.16	0.52	1.35
**Global saturated fatty acids (g)**	0.269	0.406	1.217	0.197	0.244	0.140	0.325
**Global monounsaturated fatty acids (g)**	0.2	0.445	2.178	0.208	0.149	0.161	0.432
**Polysaturated fatty acids (g)**	0.627	1.258	2.535	0.767	0.560	0.138	0.487
**Carbohydrates (g)**	71.18	70.19	66.27	75.86	77.72	79.15	18.7
**Sugars (g)**	0.41	6.82	0	0.98	0.80	0	6.26
**Dietary fibers (g)**	12.2	10.7	10.6	15.1	15.6	0	2
**Vitamin A IU**	9	10	0	11	22	0	187
**Thiamine (B1)** **(mg; % DV)**	0.383; 33%	0.364; 32%	0.763; 66%	0.316; 26%	1.191;16%	0.07; 6%	0.155; 13%
**Riboflavin (B2) (mg; % DV)**	0.115; 10%	0.113; 9%	0.139; 12%	0.251; 19%	0;0%	0.048; 4%	0.055; 4%
**Niacin (B3)** **(mg; % DV)**	5.464; 36%	6.843; 46%	0.961; 6%	4.27; 27%	4.604;29%	1600; 10%	1.77; 11%
**Pantothenic acid (B5) (mg; % DV)**	0.954; 19%	1.068; 11%	1.349; 27%	1.456; 29%	0.282;6%	1287; 26%	0.717; 14%
**Vitamin B6** **(mg; % DV)**	0.3; 23%	0.230; 18%	0.120; 9%	0.294; 23%	0.260;20%	0.171; 13%	0.093; 7%
**Folate (B9)** **(μg; % DV)**	38; 10%	45; 11%	56; 5%	38; 10%	23;6%	6; 1.5%	42; 11%
**Vitamin B12** **(μg; % DV)**	0; 0%	0; 0%	0; 0%	0; 0%	0;0%	0; 0%	0; 0%
**Vitamin E** **(mg; % DV)**	1.01; 7%	0.79; 5%	0; 0%	0.85; 6%	0.02;0%	0; 0%	0.07; 0%
**Vitamin K** **(μg; % DV)**	1.9; 2%	3.6; 3%	0; 0%	5.9; 5%	2.2;2%	0; 0%	0.3; 0%
**Calcium** **(mg; % DV)**	29; 3%	27; 3%	54; 5.58%	24; 2.48%	29;2%	1; 0.08%	2; 0.2%
**Iron (mg; % DV)**	3.19; 25%	4.44; 34%	5; 38%	2.63; 15%	2.5;14%	0.2; 1%	0.52; 3%
**Magnesium** **(mg; % DV)**	126; 35%	136; 38%	177; 50%	110; 28%	79;20%	8; 2%	37; 9%
**Manganese** **(mg; % DV)**	3.985; 190%	3; 143%	4.9; 233%	2.577; 112%	1.322;57%	0.357; 16%	0.163; 7%
**Phosphorus** **(mg; % DV)**	288; 41%	401; 57%	523; 75%	332; 47%	221;32%	33; 5%	89; 13%
**Potassium** **(mg; % DV)**	363; 8%	388; 8%	429; 9%	510; 11%	280;6%	26; 1%	270; 6%
**Selenium** **(μg; % DV)**	70.7; 129%	11.7; 17%	Not reported	13.9; 25%	37.7;69%	0; 0%	0.6; 1%
**Sodium** **(mg; % DV)**	2; 0.13%	8; 0.53%	2; 0.13%	2; 0.13%	9;0%	0; 0%	15; 1%
**Zinc (mg; % DV)**	2.65; 28%	3.28; 35%	4; 42%	2.65; 28%	2.13;19%	0.4; 4%	0.46; 4%

**Table 2 nutrients-13-02540-t002:** Poly- and monounsaturated fatty acids, serum cholesterol levels, and CVD.

Author and Year of Publication	Study Design	Duration of Study	Sample Size	Lipoprotein Levels and CVD
Mensink, 1992 [66]	Meta-analysis of 27 case-control studies	14–91 days	682 subjects, 474 men and 208 women	**Carbohydrates in the diet replaced isocaloricalry by saturated fatty acids:** Increase HDL cholesterol (*p* < 0.001), LDL cholesterol (*p* < 0.001), Total Cholesterol (*p* < 0.001); lower triglycerides (*p* < 0.001). **Carbohydrates in the diet replaced isocaloricalry by monounsaturated fatty acids:** Increase HDL cholesterol (*p* < 0.001); no effects on LDL cholesterol (*p* = 0.114), Total Cholesterol (*p* = 0.342); lower triglycerides (*p* < 0.001). **Carbohydrates in the diet replaced isocaloricalry by polyunsaturated fatty acids:** Increase HDL cholesterol (*p* = 0.002), LDL cholesterol (*p* = 0.002), Total Cholesterol (*p* < 0.001), triglycerides (*p* < 0.001).
Maki, 2017 [67]	Randomized, double-blind, crossover trial	21-day treatment (54 g per day of CO or EVOO) 21-day washout	54 volunteers, men and women	**CO intake vs. EVOO intake:** Total cholesterol = −0.37 vs. 0.02 mmol/L (*p* > 0.001); LDL = −0.36 vs. −0.08 mmol/L (*p* > 0.001); VLDL = −0.03 vs. 0.04 mmol/L (*p* > 0.001); non-HDL = −0.39 vs. −0.04 mmol/L (*p* > 0.001). ApoB = −9.0 vs. −2.5 mg/dl (*p* > 0.001). HDL = 0.02 vs. 0.05 mmol/L (*p* = 0.112). **EVOO intake vs. CO intake:** ApoA1 = 4.6 vs. 0.7 mg/dl (*p* = 0.016).
Hu, 1997 [68] Willet, 2012 [69]	Prospective Cohort Study	Follow-up: 14 years	80,082 women, from the cohort of the Nurses’ Health Study	**CHD Risk for each 5% increase in energy intake from saturated fats:** RR = 1.17; 95% CI = 0.97–1.41; *p* = 0.10. **CHD Risk for each 2% increase in energy intake from trans-unsaturated fats:** RR = 1.93; 95% CI = 1.43–2.61; *p* = 0.001) **CHD Risk for each 5% increase in energy intake from monounsaturated fats:** RR = 0.81; 95% CI = 0.65–1.00; *p* = 0.05). **CHD Risk for each 5% increase in energy intake from polyunsaturated fats:** RR = 0.62; 95% CI = 0.46–0.85; *p* < 0.003. **CHD Risk by replacing 5% energy from saturated fat with unsaturated fat:** RR = 0.58; 95% CI = 0.23–0.56; *p* < 0.001) **CHD Risk by replacing 2% of energy from trans unsaturated fat with un-hydrogenated, unsaturated fats:** RR = 0.47; 95% CI = 0.34–0.67; *p* < 0.001.
Jakobsen, 2009 [70]	Meta-analysis of prospective cohort studies	Follow-up: 4 to 10 years	344,696 subjects from 11 American and European studies included in the Pooling Project of Cohort Studies on Diet and Coronary Disease	**CHD Risk by replacing 5% of energy from SFA with MUFA or PUFA or carbohydrates (CHs):** MUFAs vs. SFAs: HR= 1.19; 95% CI = 1.00–1.42. PUFAs vs. SFAs: HR = 0.87; 95% CI = 0.77–0.97. CHs vs. SFAs: HR = 1.07; 95% CI = 1.01–1.14. **Coronary deaths Risk by replacing 5% of energy from SFA with MUFA or PUFA or carbohydrates (CHs):** MUFAs vs. SFAs: HR = 1.01; 95% CI = 0.73–1.41. PUFAs vs. SFAs: HR = 0.74; 95% CI = 0.61–0.89. CHs vs. SFAs: HR = 0.96; 95% CI = 0.82–1.13.
Lai, 2019 [71]	Prospective Cohort Study	Follow-up: 22 years	3869 subjects from the cohort of the Cardiovascular Health Study (CHS)	**Palmitic acid (16:0) and risk of mortality** All-cause mortality: HR = 1.35; 95% CI = 1.17–1.56; *p* < 0.001. CVD mortality: HR = 1.44; 95% CI = 1.18–1.76; *p* < 0.001. Non-CVD mortality: HR = 1.36; 95% CI = 1.16–1.59; *p* < 0.001. **Palmitoleic acid (16:1n-7) and risk of mortality** All-cause mortality: HR = 1.40; 95% CI = 1.21–1.62; *p* < 0.001. CVD mortality: HR = 1.42; 95% CI = 1.15–1.76; *p* = 0.001. Non-CVD mortality: HR = 1.30; 95% CI = 1.12–1.52; *p* = 0.001 **Stearic acid (18:0) and risk of mortality** All-cause mortality: HR = 0.76; 95% CI = 0.66–0.88; *p* < 0.001. CVD mortality: HR = 0.77; 95% CI = 0.62–0.94; *p* = 0.003. Non-CVD mortality: HR = 0.72; 95% CI = 0.62–0.84; *p* < 0.001 **Oleic acid (18:1n-9) and risk of mortality** All-cause mortality: HR = 1.56; 95% CI = 1.35–1.80; *p* < 0.001. CVD mortality: HR = 1.48; 95% CI = 1.21–1.82; *p* < 0.001. Non-CVD mortality: HR = 1.50 95% CI = 1.28–1.75; *p* < 0.001. **Palmitic acid (16:0) and risk of incident CVD** Fatal and non-fatal CVD: HR = 1.20; 95% CI = 1.01–1.43; *p* = 0.029. Fatal and non-fatal CHD: HR = 1.13; 95% CI = 0.93–1.38; *p* = 0.287. Fatal and non-fatal Stroke: HR = 1.26; 95% CI = 0.96–1.66; *p* = 0.028. **Palmitoleic acid (16:1n-7) and risk of incident CVD** Fatal and non-fatal CVD: HR = 1.28; 95% CI = 1.07–1.53; *p* = 0.012. Fatal and non-fatal CHD: HR = 1.07; 95% CI = 0.88–1.31; *p* = 0.506. Fatal and non-fatal Stroke: HR = 1.38; 95% CI = 1.05–1.83; *p* = 0.038. **Stearic acid (18:0) and risk of incident CVD** Fatal and non-fatal CVD: HR = 0.82; 95% CI = 0.69–0.97; *p* = 0.003. Fatal and non-fatal CHD: HR = 0.93; 95% CI = 0.77–1.13; *p* = 0.266. Fatal and non-fatal Stroke: HR = 0.77; 95% CI = 0.59–1.00; *p* = 0.013. **Oleic acid (18:1n-9) and risk of incident CVD** Fatal and non-fatal CVD: HR = 1.33; 95% CI = 1.12–1.57; *p* < 0.001. Fatal and non-fatal CHD: HR = 1.23; 95% CI = 1.01–1.48; *p* = 0.008. Fatal and non-fatal Stroke: HR = 1.34; 95% CI = 1.02–1.75; *p* = 0.005.
Borges, 2020 [73]	Meta-analysis of prospective cohort and case-control studies	Follow-up: 10 to 25 years	23,518 subjects from 5 cohort studies and 1 case-control study, from the UCL-LSHTM-Edinburgh-Bristol (UCLEB) Consortium	**DHA and risk for CHD** OR = 0.85; 95% CI = 0.76–0.95 **LA and risk for CHD** OR = 1.01; 95% CI = 0.87–1.18 **MUFA and risk for CHD** OR = 1.36; 95% CI = 1.15–1.61 **SFA and risk for CHD** OR = 0.94; 95% CI = 0.82–1.09 **DHA and risk for Stroke** OR = 0.95; 95% CI = 0.89–1.02 **LA and risk for Stroke** OR = 0.82; 95% CI = 0.75–0.90 **MUFA and risk for Stroke** OR = 1.22; 95% CI = 1.03–1.44 **SFA and risk for Stroke** OR = 0.94; 95% CI = 0.79–1.11
Lee, 2020 [74]	Prospective Cohort Study	Follow-up: 22 years	4249 subjects from the cohort of the Cardiovascular Health Study (CHS)	**Habitual levels of plasma fatty acids and risk of incident HF** palmitic acid: HR = 1.17, 95% CI 1.00–1.36; 7-hexadecenoic acid: HR = 1.05, 95% CI 0.92–1.18; vaccenic acid: HR = 1.06, 95% CI 0.92–1.22; but changes in levels were associated with a higher risk of HF (HR = 1.43, 95% CI 1.18–1.72); myristic acid: HR = 0.90, 95% CI = 0.77–1.05; palmitoleic acid: HR = 1.01, 95% CI = 0.88–1.16; stearic acid: HR = 0.94, 95% CI = 0.81–1.09; oleic acid: HR = 1.13, 95% CI = 0.98–1.30; **Change in serial levels of plasma fatty acids and risk of incident HF** palmitic acid: HR = 1.26 95% CI 1.03–1.55; 7-hexadecenoic acid: HR = 1.36, 95% CI 1.13–1.62; vaccenic acid: HR = 1.43, 95% CI 1.18–1.72; myristic acid: HR = 1.11, 95% CI = 0.91–1.36; palmitoleic acid: HR = 1.06, 95% CI = 0.87–1.28; stearic acid: HR = 0.94, 95% CI = 0.76–1.15; oleic acid: HR = 1.13, 95% CI = 0.93–1.37.

CO: corn oil; EVOO: extra virgin olive oil; CHD: coronary heart disease; LA: linoleic acid; HF: heart failure

**Table 3 nutrients-13-02540-t003:** Diet pattern and risk of frailty, cardiovascular risk, and mortality.

Author and Year of Publication	Study Design	Duration of Study	Sample Size	Risk of Frailty and Mortality
Lo, 2017 [86]	Cross-sectional study	3 years	923 subjects aged 65 years and older from the cohort of Nutrition and Health Survey in Taiwan (NAHSIT)	**Associations between tertiles of dietary pattern score and frailty according Fried criteria:** OR = 0.12 (95% CI = 0.02–0.76; *p* = 0.019) for tertile 3 of dietary pattern score. **Associations between tertiles of dietary pattern score and pre-frailty according Fried criteria:** OR = 0.40 (95% CI = 0.19–0.83; *p* = 0.015) for tertile 3 of dietary pattern scores.
Trichopoulou, 2003 [6]	Population-based, prospective study	Median duration of follow-up: 3.7 years	8895 men and 13,148 women	**All-cause death:** HR = 0.75 (95% CI 0.64–0.87) for a Two-Point Increase in the Mediterranean-Diet Score **Death from CHD** (coronary heart disease): HR = 0.67 (95% CI 0.47–0.94) for a Two-Point Increase in the Mediterranean-Diet Score **Death from cancer:** HR = 0.76 (95% CI 0.59–0.98) for a Two-Point Increase in the Mediterranean-Diet Score
Estruch, 2018 [9]	Parallel-group, multicenter, randomized trial	Median duration of follow-up: 4.8 years	1050 men and 1493 women with MD(Mediterranean-Diet) with EVOO(extra virgin olive oil) 1128 men and 1326 women with MD with nuts 987 men and 1463 women with Control Diet	**Myocardial infarction:** HR = 0.82 (95% CI 0.52–1.30) for MD with EVOO vs. Control Diet HR = 0.76 (95% CI 0.47–1.25) for MD with Nuts vs. Control Diet **Stroke:** HR = 0.65 (95% CI 0.44–0.95) for MD with EVOO vs. Control Diet HR = 0.54 (95% CI 0.35–0.82) for MD with Nuts vs. Control Diet **Death from CVD:** HR = 0.62 (95% CI 0.36–1.06) for MD with EVOO vs. Control Diet HR = 1.02 (95% CI 0.63–1.67) for MD with Nuts vs. Control Diet **All-cause death:** HR = 0.90 (95% CI 0.69–1.18) for MD with EVOO vs. Control Diet HR = 1.12 (95% CI 0.86–1.47) for MD with Nuts vs. Control Diet
Sofi, 2008 [10]	Meta-analysis of prospective cohort studies	Follow-up time range: from 3.7 to 18 years	1,574,299 subjects from 12 studies	**Mortality from CVD:** RR = 0.91 (95% CI 0.87–0.95) **All-cause mortality:** RR = 0.91 (95% CI 0.89–0.94 **Mortality from cancer:** RR = 0.94 (95% CI 0.92–0.96) **Incidence of Parkinson’s disease and Alzheimer’s disease:** RR = 0.87 (95% CI 0.80–0.96)
Sofi, 2010 [11]	Meta-analysis of prospective cohort studies	Follow-up time range: from 4 to 20 years	508,393 subjects from 7 studies	**Mortality from CVD:** RR = 0.90 (95% CI 0.87–0.93) **All-cause mortality:** RR = 0.92 (95% CI 0.90–0.94) **Mortality from cancer:** RR = 0.94 (95% CI 0.92–0.96) **Incidence of neurodegenerative disease:** RR = 0.87 (95% CI 0.81–0.94)
Kromhout, 2018 [89]	Prospective Cohort Study	Follow-up time: 50-years	12,763 subjects from 16 cohorts of the Seven Countries Study.	**Mortality from CVD:** Inverse correlation between consumption of cereals, vegetables, legumes, and alcohol and long-term CHD mortality rates (r = −0.52 to −0.62) Direct correlation between consumption of hard fat plus sweet products, animal foods except fish, and long-term CHD mortality rates (r = 0.68 to 0.84)
Zaslavsky, 2018 [90]	Prospective Cohort Study	Mean follow-up: 12.4 years	10,431 women aged 65–84 year from the cohorts of the Women’s Health Initiative Observational Study	**Associations between of dietary pattern and mortality:** HR = 0.91, 95% CI: 0.84–0.99, *p* = 0.02, for high intake of vegetables; HR = 0.87, 95% CI: 0.80–0.94, *p* < 0.001, for high intake of nuts; HR = 0.83, 95% CI: 0.77–0.90, *p* < 0.001, for high intake of whole grains.
Campanella, 2020 [95]	Prospective Cohort Study	Median follow-up time: 12.82, 12.91 and 12.84 years for high, medium and low rMED subjects	5152 subjects from the cohorts of MICOL/PANEL and NUTRIHEP Study (2851 from MICOL/PANEL; 2301 from NUTRIHEP)	**Associations between of dietary pattern and mortality:** Direct correlation between higher adherence to the MD at baseline and mortality. Higher adherence to the MD at baseline was related to a lifespan 6.21 and 8.28 years longer.
Hernaez, 2019 [100]	Parallel-group, multicenter, randomized trial	Follow-up time: 1 year.	296 subjects from the cohort of the PREDIMED Study	**Association among food groups and improvements in HDL functions:** Increments in cholesterol efflux capacity: +0.7% (*p* = 0.026) for increase in daily intake of 10 g of EVOO; +0.6% (*p* = 0.017) for increase in daily intake of 25 g of WG; –1.1% (*p* = 0.010) for increase in daily intake of 25 g of fish. Increments in PON1(Paraoxonase 1) activity: +12.2% (*p* = 0.049) for increase in daily intake of 30 g of nuts; +11.7% (*p* = 0.043) for increase in daily intake of 25 g of legume; +3.9% (*p* = 0.030) for increase in daily intake of 25 g of fish. Decreases in CETP(cholesteryl ester transfer protein) activity: –4.8% (*p* = 0.028) for increase in daily intake of 25 g of legume; –1.6%, (*p* = 0.021) for increase in daily intake of 25 g of fish.

**Table 4 nutrients-13-02540-t004:** Whole grains(WG) intake, cardiovascular risk factors and body weight.

Author and Year of Publication	Study Design	Duration of Study	Sample Size	Effect of WG Intake on Cardiovascular Risk Factors and Body Weight
Kelly, 2017 [110]	Meta-analysis of RCTs	Duration of studies: 12 to 16 weeks	1414 subjects from 9 RCTs	**Total CVD mortality and CVD events:** Authors did not find any studies that reported significative effects of WG foods on total cardiovascular mortality or cardiovascular events. **CVD risk factors (mean difference, MD; 95% CI):** Body weight change (kg) = (MD −0.41; 95% CI = −1.04–0.23); BMI = (MD −0.12; 95% CI = −0.24–0.01); Total cholesterol (mmol/L) = (MD 0.07; 95% CI = −0.07–0.21); LDL cholesterol (mmol/L) = (MD 0.06; 95% CI = −0.05–0.16); HDL cholesterol (mmol/L) = (MD −0.02; 95% CI = −0.05–0.01); Triglycerides (mmol/L) = (MD 0.03; 95% CI = −0.08–0.13); SBP(systolic blood pressure) (mmHg) (MD 0.04; 95% CI = −1.67–1.75); DBP(diastolic blood pressure) (mmHg) (MD 0.16; 95% CI = −0.89–1.21).
Kirwan, 2016 [111]	Double-blind, randomized, controlled crossover study	Duration of study: 8 weeks, with a 10 weeks washout period between diets	33 overweight or obese men and women.	**Body weight:** No significant difference between WG vs. control diets. **SBP:** No significant difference between WG vs. control diets (*p* = 0.80). **DBP:** WG vs. control diet = (−5.8 mm Hg (95% CI = 27.7–24.0) vs. −1.6 mm Hg (95% CI = 24.4–1.3 mm Hg), *p* = 0.01. **Total Cholesterol and LDL Cholesterol:** No significant difference between WG vs. control diets **HbA1c** (glycated hemoglobin): WG diet significantly lowered HbA1c (*p* = 0.04) **FPI** (fasting plasma insulin): WG diet significantly lowered FPI (*p* = 0.02) **Adiponectin:** WG vs. control diet = −0.1 mg/mL (95% CI = −0.9–0.7) vs. −1.4 mg/mL (95% CI = −2.6–−0.3), *p* = 0.05.
Marventano, 2017 [112]	Meta-analysis of RCTs	Where available, AUC(area under the curve) values range from 0 to 240 min	206 subjects from 14 RCTs	**Changes from baseline in glucose iAUC values at 120 min (MD; 95% CI):** MD = −29.71 mmol x min/L; 95% CI = −43.57–−15.85 **Changes from baseline in insulin iAUC values at 120min (MD; 95% CI):** MD = −2.01 nmol x min/L; 95% CI = −2.88–−1.14
Musa-Veloso, 2018 [114]	Meta-analysis of RCTs	Where available, AUC values range from 0 to 120 min	274 subjects from 20 RCTs	**Postprandial blood glucose AUC of WG vs. refined wheat, rice, or rye:** WG vs. white wheat: AUC = −6.7 mmol/L x min; 95% CI = −25.1–11.7; *p* = 0.477. WG vs. endosperm rye: AUC = −5.5 mmol/L x min; 95% CI = −24.8–13.8; *p* = 0.576. WG vs. white rice: AUC = −40.5 mmol/L x min; 95% CI = −59.6–−21.3; *p* < 0.001.
Kirø, 2018 [115]	Prospective Cohort Study	Median follow-up: 15 years	55,565 subjects (26,251 men, 29,214 women) from the Diet, Cancer, and Health Cohort	**Increment of 16 g/day of WG intake and risk of type 2 diabetes:** Men: HR = 0.89, 95% CI = 0.87, 0.91 Women: HR = 0.93, 95% CI = 0.91. 0.96 **Highest vs. lowest quartile of WG intake and risk of type 2 diabetes:** Men: HR = 0.66, 95% CI: 0.60–0.72, *p* < 0.0001 Women: HR = 0.78, 95% CI: 0.70–0.86, *p* < 0.0001 **Increment of 50 g/day of WG intake and risk of type 2 diabetes:** Men: HR = 0.88, 95% CI = 0.86–0.90 Women: HR = HR = 0.93, 95% CI = 0.90–0.96 **Highest vs. lowest quartile of WG intake and risk of type 2 diabetes:** Men: HR = 0.63, 95% CI: 0.58–0.69, *p* < 0.0001 Women: HR = 0.80, 95% CI: 0.72–0.88, *p* < 0.0001
Maki, 2019 [117]	Meta-analysis of observational studies and RCTs	Mean duration of 3 prospective cohort studies: 8 years. Mean duration of 9 cross-sectional studies: 5 years. Mean duration of 9 RCTs: 90 days	136,834 subjects from 12 observational studies (3 prospective cohort studies and 9 cross-sectional studies) and 973 subjects from 9 RCTs	**Meta-Regression Analysis from Cross-Sectional Studies:** Inverse correlation between WG consumption and BMI (r = −0.526, *p* = 0.0001) **Qualitative Analysis from Prospective Cohort Studies:** Inverse association between WG consumption and weight change, with a follow-up period from 5 to 20 years **Meta-Regression of RCTs:** No significant difference between WG consumption and weight change (standardized MD = −0.049 Kg; 95% CI = −0.388–0.199; *p* = 0.698)

**Table 5 nutrients-13-02540-t005:** Whole grains(WG) intake and reduction of mortality.

Author and Year of Publication	Study Design	Duration of Study	Sample Size	Highest vs. Lowest Whole Grains Intake and Reduction of Mortality
Ma, 2016 [118]	Meta-analysis of prospective cohort studies	Median follow-up time: 5.9 to 26 years	809,901 subjects (99,224 deaths) from 10 prospective cohort studies	**WG intake and all-cause mortality:** RR = 0.82; 95% CI = 0.78–0.87 **Increment of 1 serving/day of WG intake and all-cause mortality risk:** RR = 0.93; 95% CI = 0.89–0.97
Zong, 2016 [119]	Meta-analysis of prospective cohort studies	Median follow-up time: 6 to 28 years	786,076 subjects (97,867 deaths) form 14 prospective cohort studies	**WG intake and all-cause death:** RR = 0.84; 95% CI = 0.80–0.88; *p* < 0.001 **WG intake and death from CVD:** RR = 0.82; 95% CI = 0.79–0.85; *p* < 0.001 **WG intake and death from cancer:** RR = 0.88; 95% CI = 0.83–0.94; *p* < 0.001 **Increment of 1 serving/day of WG intake and total mortality risk:** RR = 0.93; 95% CI = 0.92–0.94 **Increment of 1 serving/day of WG intake and CVD mortality risk:** RR = 0.91; 95% CI = 0.90–0.93 **Increment of 1 serving/day of WG intake and cancer mortality risk:** RR = 0.95; 95% CI = 0.94–0.96
Wei, 2016 [120]	Meta-analysis of prospective cohort studies	Median follow-up time: 14 years (range: 5.5–26 years)	816,599 subjects (89,251 deaths) form 11 prospective cohort studies	**WG intake and all-cause death:** SRR = 0.87; 95% CI = 0.84–0.90 **WG intake and death from CVD:** SRR = 0.81; 95% CI = 0.75 – 0.89 **WG intake and death from cancer:** SRR = 0.89; 95% CI = 0.82 – 0.96 **Increment of 3 serving/day of WG intake and total mortality risk:** SRR = 0.81; 95% CI = 0.76 – 0.85 **Increment of 3 serving/day of WG intake and CVD mortality risk:** SRR = 0.74; 95% CI = 0.66 – 0.83 **Increment of 3 serving/day of WG intake and cancer mortality risk:** SRR = 0.91; 95% CI = 0.84 – 0.98
Aune, 2016 [36]	Meta-analysis of prospective cohort studies	Follow-up time range: 3–26 years	245,012 to 705,253 subjects (34,346 deaths from cancer; 100,726 deaths from any cause) from 45 prospective studies	**WG intake and death from CHD:** RR = 0.65; 95% CI = 0.52–0.83 **WG intake and death from Stroke:** RR = 0.85; 95% CI = 0.64–1.13 **WG intake and death from CVD:** RR = 0.81; 95% CI = 0.75–0.87 **WG intake and death from cancer:** RR = 0.89; 95% CI = 0.82–0.96 **WG intake and all-cause death:** RR = 0.82; 95% CI = 0.77–0.88 **Increment of 3 serving/day of WG intake and total CHD mortality risk:** RR = 0.81; 95% CI = 0.74–0.89 **Increment of 3 serving/day of WG intake and Stroke mortality risk:** RR = 0.86; 95% CI = 0.74–0.99 **Increment of 3 serving/day of WG intake and CVD mortality risk:** RR = 0.71; 95% CI = 0.61–0.82 **Increment of 3 serving/day of WG intake and cancer mortality risk:** RR = 0.85; 95% CI = 0.80–0.91 **Increment of 3 serving/day of WG intake and all-cause mortality risk:** RR = 0.83; 95% CI = 0.77–0.90
Benisi-Kohansal, 2016 [121]	Meta-analysis of prospective cohort studies	Follow-up time range: 5.5–26 years	2,282,603 subjects from 20 prospective cohort studies	**WG intake and all-cause death:** RR = 0.87; 95% CI = 0.84–0.91 **WG intake and death from CVD:** RR = 0.84; 95% CI = 0.78–0.89 **WG intake and death from cancer:** RR = 0.94; 95% CI = 0.91, 0.98 **Increment of 3 serving/day of WG intake and total all-cause mortality risk:** SRR = 0.83; 95% CI = 0.79–0.88 **Increment of 3 serving/day of WG intake and CVD mortality risk:** SRR = 0.75; 95% CI = 0.68–0.83 **Increment of 3 serving/day of WG intake and cancer mortality risk:** SRR = 0.90; 95% CI = 0.83–0.98
Zhang, 2018 [122]	Meta-analysis of prospective cohort studies	Follow-up time range: 4–26 years	1,041,692 subjects (96,710 deaths) from 19 prospective cohort studies	**WG intake and all-cause death:** RR = 0.84; 95% CI = 0.81–0.88 **WG intake and death from CVD:** RR = 0.83; 95% CI = 0.79–0.86 **WG intake and death from cancer:** RR = 0.94; 95% CI = 0.87–1.01 **Increment of 1 serving/day of WG intake and total all-cause mortality risk:** RR = 0.91; 95% CI = 0.90–0.93 **Increment of 1 serving/day of WG intake and CVD mortality risk:** RR = 0.86; 95% CI = 0.83–0.89 **Increment of 1 serving/day of WG intake and cancer mortality risk:** RR = 0.97; 95% CI = 0.95–0.99

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
