# Peer review of "Whole-Grain Intake in the Mediterranean Diet and a Low Protein to Carbohydrates Ratio Can Help to Reduce Mortality from Cardiovascular Disease, Slow Down the Progression of Aging, and to Improve Lifespan: A Review"

_nutrients, 2021, doi:10.3390/nu13082540_

Round 1

Reviewer 1 Report

Despite most of my comment have been correctly addresses, and the manuscript has been improved, I still think point 4 should be fixed.

  1. In general, the link between amount of WG and duration of diet is not clear. Possible mechanism of WG effect is not included. What is the effect of WG on the molecular hallmark of aging?. Can non-pharmacological approaches (such exercise) that improve longevity and decrease risk of chronic age-associated diseases increase the effect of WG?

Author reply:

The different molecular mechanisms hypothesized are described in the manuscript. 

This point is not clear as there are mechanisms in different part of the manuscript. I suggest add a figure summarizing the main mechanisms included in the manuscript.

Reviewer 2 Report

I think author's changes are now substantial for its publication. However I did not perform any plagiarism checking test. I consider that this activity must be performed by journals previously to the acceptance for submission. Nonetheless I trust in author's statement about the verification of plagiarism check

Author Response

Thank you very much for your observation. Really, I performed a plagiarism check. I found no more than 5% similarities in the whole text and no more than  3% similarities in the tables. 

This manuscript is a resubmission of an earlier submission. The following is a list of the peer review reports and author responses from that submission.

Round 1

Reviewer 1 Report

The current manuscript entitled “Whole grain intake in the Mediterranean Diet and a low protein to carbohydrates ratio can help to slow down the progression of aging and to improve the lifespan: a review” written by Prof. Cristiano Capurso highlights the role of Mediterranean Diet (MD) in longevity and lifespan, focused on the benefits of Whole grain intake as main contributor for low protein carbohydrates ratio in the diet. This topic is very interesting because is in fashion, specially with the current controversial use of high-protein enriched diet to weight loss. However, despite the interesting content in this review, I have some concerns that makes me to decline its acceptance, considering that some unethical practice could be carriying out in the writing process of the manuscript. The review contains sentences extracted from other published works which is infringing the plagiarism rules. Here are some examples:

The sentence “…Between 2015 and 2030, the number of people in the world 27 aged 60 years or over is projected to grow by 56 per cent, from 901 million to 1.4 billion, 28 and by 2050, the global population of older persons is projected to more than double its 29 size in 2015, reaching nearly 2.1 billion…” it can be found on https://www8.hp.com/us/en/hp-information/accessibility-aging/aging megatrend.html#:~:text=56%25%20population%20increase,2015%2C%20reaching%20nearly%202.1%20billion.

The sentence “Over the next 15 years, the number of 33 older persons is expected to grow fastest in Latin America and the Caribbean with a projected 71 per cent increase in the population aged 60 years or over, followed by Asia (66 35 per cent), Africa (64 per cent), Oceania (47 per cent), Northern America (41 per cent) and 36 Europe (23 per cent)” is literally the same sentence that appears in the Key trends in population ageing for the 2030 Agenda for Sustainable Development

The sentence “These comprise about 45−50% cell wall material [30] with the pericarp, which is the major 145 tissue, being more similar in cell wall composition to wheat straw than to other seed tissues, with about 30% cellulose, 60% arabinoxylan, and 12% lignin [31].” appears in Shewry et al. 2013.

The sentence “The most abundant phenolic acids in wheat are derivatives of hydroxycinnamic acid, overo dehydrodimers and dehydrotrimers of ferulic acid and  synapic and p-coumaric acids” is from Laddomada et al., 2015

And “In addition, wheat is a rich source of glycine betaine, with smaller amounts of choline (the precursor of betaine) and trigonelline (a structural analogue of betaine and choline). These are all called "methyl donors"… Wheat is an important dietary source of B vitamins, particularly thiamin (B1), riboflavin (B2), niacin (B3), pyridoxine (B6), and folates (B9)” is from Shewry et al. 2013.

These examples were found by simple randomly checking in google. Using Fix Plagiarism Fast software, the manuscript did not past the plagiarism test either.

I would be glad to review the manuscript again if the author rewrite the manuscript and pass the plagiarism test. But it must be Editor’s decision to allow the manuscript resubmission

Other some concerns about the content of the review are:
The author is missing the role of barley as one of the main grain cultures in the Mediterranean diet since before the beginnings of the written history. However Maize, which culture is originated from America and introduced in Europe in late XV century has a space in the review. I encourage the author to include barley or give the reasons why barley is excluded from the review introduction.
In my opinion, the topic related with the “Reduction of protein to carbohydrates ratio influence aging and lifespan” is underdeveloped respect to the other parts, specially considering the title of the review. I strongly recommend increase the content of this part, even (it was necessary) in detrimental of other content like the description of the different grains or diet pattern, which have been widely described in other review manuscripts.

As minor suggestions, I would remove the definitions of Degree of Polymerization and Dietary Fibers from the Codex definition and Commission directive 2008/100/EC. I think they can be referenced but not necessarily described. I think that information is not relevant for the review comprehension. In addition, I also recommend increase the size (or bold) the letters in the pictures and also double-check misspelling mistakes.

Author Response

Thank you for your comments. I checked the entire manuscript and checked the bibliographic citations for plagiarism check. 

I also included the barley in the manuscript.

I have expanded the content of the section concerning the protein-carbohydrate ratio. 

I removed the definitions of Degree of Polymerization and Dietary Fibers from the Codex definition and Commission directive 2008/100/EC. 

About the size and bold of the letters, I followed the format present in MDPI

Reviewer 2 Report

Overall, this is an interesting manuscript, as the authors summarizing the impact of whole grains (WG) and derivatives, in the diet, and low protein-carbohydrate ratio on aging progression, mortality and lifespan. However, my main concern is that appropriated discussion and author point-of-view are not clear. Moreover, variability in diet and genetics between studies, as well as  control of physical activity and lifestyle made the role of WG on longevity and chronic diseases management unclear.

  1. Cardiovascular disease is an important issue in the manuscript, however it is not included in the title.
  2. The organization of the manuscript is not clear. Discussion section should be included.
  3. In general, I think it is important that reader understand the author point-of-view. I suggest include a paragraph in each section of the manuscript with the author- point of view
  4. In general, the link between amount of WG and duration of diet is not clear. Possible mechanism of WG effect is not included. What is the effect of WG on the molecular hallmark of aging?. Can non-pharmacological approaches (such exercise) that improve longevity and decrease risk of chronic age-associated diseases increase the effect of WG?
  5. Some information about duration of each study should be included in the tables.
  6. New table of the section 2.6.4. Poly- and monounsaturated fatty acids, serum cholesterol levels and cardiovascular disease will help to understand.
  7. Conclusion section is not clear. I recommend rewrite it and include specific recommendation.
  8. EXTENSIVE editing of English language and style required

Author Response

Thank you for your comments and suggestions:

1. I changed the title of the manuscript as follows: 

Whole-grain intake in the Mediterranean Diet and a low protein to carbohydrates ratio can help to reduce mortality from cardiovascular disease, slow down the progression of aging, and to improve lifespan: a review.

2. I included a discussion section. 

3.  I included a paragraph in each section of the manuscript with the author- point of view.

4. Actually, the intake of WG is an integral part of a diet to be followed throughout life as part of a healthy lifestyle, which also includes physical exercise. The different molecular mechanisms hypothesized are described in the manuscript. 

5. I included in all the tables information about the duration of each study.

6. I have inserted an additional table in section 2.6.4.

7. I rewrote the Conclusion section, which includes some specific recommendations.

8. I have completely revised the manuscript bringing the appropriate corrections to the English form.